



# Quantifying Bioaerosol Concentrations in Dust Clouds through Online UV-LIF and Mass Spectrometry Measurements at the Cape Verde Atmospheric Observatory

Douglas Morrison,[1] Ian Crawford,[1] Nicholas Marsden,[1] Michael Flynn,[1] Katie Read,[2] Luis Neves,[2] Virginia Foot,[4] Paul Kaye,[3] Warren Stanley,[3] Hugh Coe,[1] David Topping,[1] and Martin Gallagher.[1]

[1]Department of Earth and Environmental Science, University of Manchester, Brunswick St, Manchester, M13 9PS
[2]Wolfson Atmospheric Chemistry Laboratory, University of York, York, YO10 5DD
[3]Science and Technology Research Institute, University of Hertfordshire, Hatfield, U.K.
[4]Defence Science and Technology Laboratory, Salisbury, U.K

**Correspondence:** Douglas Morrison (douglas.morrison@manchester.ac.uk)

**Abstract.** Observations of the long-range transport of biological particles in the tropics via dust vectors are now seen as fundamental to the understanding of many global atmosphere-oceanic biogeochemical cycles, changes in air quality, human health, ecosystem impacts, and climate. However, there is a lack of long-term measurements quantifying their presence in such conditions. Here we present annual observations of bioaerosol concentrations based on online ultraviolet light induced fluorescence

(UV-LIF) spectrometry from the global WMO/Global Atmospheric Watch (GAW) observatory on Sao Vicente Cape Verde Atmospheric Observatory. We observe the expected strong seasonal changes in absolute concentrations of bioaerosols with significant enhancements during winter due to the strong island inflow of airmass, originating from the African continent. Monthly median bioaerosol concentrations as high as $45L^{-1}$ were found with 95th percentile values exceeding $130L^{-1}$ during strong dust events. However, in contrast the relative fraction of bioaerosol numbers compared to total dust number concentration

shows little seasonal variation. Mean bioaerosol contributions accounted for 0.4±0.2% of total coarse aerosol concentrations, only rarely exceeding 1% during particularly strong events under appropriate conditions. Although enhancements in the median bioaerosol fraction do occur in winter, they also occur at other times of the year, likely due to the enhanced Aeolian activity driving dust events at this time from different sources. We hypothesise that this indicates the relative contribution of bioaerosol material in dust transported across the tropical Atlantic throughout the year is relatively uniform, comprised mainly of mixtures

of dust and bacteria and/or bacterial fragments. We argue that this hypothesis is supported from analysis of measurements also at Cape Verde just prior to the long-term monitoring experiment where UV-LIF single particle measurements were compared with Laser Ablation Aerosol Particle Time of Flight mass spectrometer (LAAP-ToF) measurements. These clearly show a very high correlation between particles with mixed bio-silicate mass spectral signatures and UV-LIF bio-fluorescent signatures suggesting the bioaerosol concentrations are dominated by these mixtures. These observations should assist with constraining

bioaerosol concentrations for tropical Global Climate Model (GCM) simulations. Note here we use the term "bioaerosol" to include mixtures of dust and bacterial material.





## 1 Introduction

Aerosols play a key role in the global climate. Their suspension in the air can scatter incoming solar radiation, reducing the warming effect of the sun. Rather than scatter light directly, aerosols can also serve as nuclei for cloud droplets, ice crystals and

precipitation. This can promote further changes in the climate, with cloud cover increasing local albedo effects. Although most cloud condensation nuclei (CCN) are inorganic, there has been increasing evidence to suggest that biological particles play an important role too. This is because despite existing in relatively low concentrations, bioaerosols are more effective CCN than alternative particle types. Bacteria such as *Pseudomonas syringae* are thought to promote rainfall (Sands et al., 1982), while strains such as *Erwinia carotovora carotocora* and *Ewinia carotovora atroseptica* have been shown to be CCN active, with

25-30% activatable at supersaturations larger than 1% (Franc and DeMott, 1998). Bauer et al. (2003) collected cultivable bacteria from cloud water samples in Austria and found all samples to be activatable at supersaturations where insoluble wettable particles of comparable size would not have been. Other biogenic aerosols such as pollen have also been shown to have CCN properties (Pope, 2010). This is also true for fragmented pollen grains, significantly raising the number of potential activation sites (Steiner et al., 2015).

Homogeneous ice nucleation occurs in liquid particles at temperatures of $-36°C$ and below. However, heterogeneous ice nucleation can occur at significantly warmer temperatures, with ice nuclei (IN) reducing the energy required for crystallisation to begin. The effectiveness of IN in this regard depends on their composition, size, surface area and more (Hoose and Möhler, 2012). They also encompass a broad range of particles, including mineral dust, metals, soot, and biological particles. Of these

particle types, mineral dust has been one of the most closely investigated, with multiple studies observing a range of conditions at which it can act as effective nuclei. Over the dust belt as outlined by Liu et al. (2008), (Zhang et al., 2012) found dust to initiate freezing in midlevel supercooled stratiform clouds (MSSCs) at temperatures of $-10°C$ and below. This is warmer than findings from Ansmann et al. (2008), who did not find evidence of ice formation in supercooled stratiform clouds with cloud-top temperatures warmer than $-18°C$; but colder than findings by Sassen et al. (2003), who attributed African dust to

the glaciation of altocumulus clouds at $-5°C$.

More recent focus has been placed on biological particles, with many indentified as more efficient IN. For example, strains of *Pseudomonas syringae* have been found to initiate freezing at temperatures as high as $-2°C$ (Yankofsky et al., 1981). They are also capable of influencing clouds across a range of altitudes, with fungal spores as large as $15um$ reaching altitudes of

6km in a matter of days (Haga et al., 2013). Dispute over bioaerosol's contributions to these meteorological processes focus on their low concentrations, with bioaeorosols often accounting for less than 1% of total particle concentrations. Some estimates are even lower, with Bauer et al. (2002) finding bacterial average mass concentrations accounted for just 0.01% of organic carbon (OC) in cloud water and precipitation samples. However, in pristine environments bioaerosol concentrations can be





significantly amplified. For example, bioaerosols have been shown to account for 46% of total coarse aerosol mass concentration in central Amazonian rainforest (Huffman et al., 2012). In these instances it is thought that bioaerosols are part of a self-sustaining cycle, promoting rainfall that in turn drives vegetative growth (Fröhlich-Nowoisky et al., 2016). Furthermore, Pratt et al. (2009) identified 33% of the ice crystal residual particles sampled in a wave cloud over Wyoming as biogenic.

Bioaerosols are also capable of global dispersion (McTainsh, 1996). This is because although some are freely suspended in the air, others can utilise mineral dust to act as a vector for their transport. It is perhaps because of this that dust clouds often contain significant concentrations of micro-organisms (Griffin et al., 2001). Furthermore, Yamaguchi et al. (2012) investigated the bacteria's ability to grow and reproduce when attached to dust, and found many to remain physiologically active. This
can have important implications for cloud formation but also ecological and health impacts too. Asian dust particles and their corresponding microbial concentrations have been found to positively correlate with levels of ice nucleation in snow samples atop Mt Tateyama (Maki et al., 2018). Meanwhile, there is a known association between dust storms and meningitis. Outbreaks often occur in the Sahelian region of North Africa between February and May, affecting as many as 200,000 individuals annually (Sultan et al., 2005). During this period conditions are usually dry and dust storms are frequent. Meningitis outbreaks
persist up until the wet season begins, when dust events cease (Molesworth et al., 2003).

Trade winds are an important aspect of Aeolian events in Africa, with the point of their convergence known as the Intertropical Convergence Zone (ITCZ). Within this zone air is made to rise, forming the ascending part of the Hadley cell. The ITCZ is also subject to latitudinal movement, following the progression of the annual solar maximum (Folland et al., 1991). This results
in a cycle of atmospheric loading, with dust readily taken to high altitudes during summer. This material is consequently often transported significant distances across the Atlantic, where it is deposited in the ocean along the way. As such, trade winds and dust events can act as major contributors for nutrient deposition. Although the chemical content of what is deposited depends on the source of the dust, most dust particles originate from arid or semi-arid regions and contain large amounts of biogeochemically significant elements. The role of organic components in affecting ocean life is coming under increased scrutiny.
For example, the decline in Caribbean coral reefs has been partially attributed to transatlantic dust events (Rypien, 2008). *Aspergillus sydowii* is a fungal spore implicated in a Caribbean-wide seafan disease that has been cultured from Caribbean air samples. It is believed that dust is acting as a substrate for said spores. With Yamaguchi et al. (2012) having analysed the bacterial community structure for Asian dust particles, it was found that more than 20 bacterial classes were present, with *Actinobacteria*, *Bacilli* and *Sphingobacteria* dominating. These findings are similar to previous studies on African dust (Griffin
et al., 2001), suggesting that the diversity of such communities is largely similar across arid regions.

Research has shown how bioaerosols can influence the world around them and has also observed their presence during dust events. However, their concentrations have not been readily quantified in such conditions. This study aimed to capture long-term trends in bioaerosol concentrations within a region of the world where dust events are both common and pronounced.
To do this, online measurements using a Wideband Integrated Bioaerosol Sensor (WIBS-4M) were taken for 11 months off



the west coast of continental Africa. This was preceded by an intensive campaign that paired a [UV-LIF] WIBS-4, e.g. Gabey et al. (2010), Savage et al. (2017), with a Laser Ablation Aerosol Time of Flight Mass Spectrometer (LAAP-ToF), Marsden et al. (2018). The latter was used to provide detailed contemporary particle composition analysis and to identify bio-silicate particle classes. This bio-silicate classification was then used to validate the WIBS-4A bio-fluorescent classification schemes,

e.g. Ruske et al. (2017), to subsequently interpret the long-term ambient data sets collected by a WIBS-4 at the CVAO site. With an estimated 50% of global annual dust originating from North Africa (Engelstaedter et al., 2006), the African continent is a high priority region that is often overlooked for such studies.

## 2 Methodology

### 2.1 Site and Long-Term Sampling Details

From September 2015 to August 2016 a WIBS-4M was continuously sampling at the Cape Verde Atmospheric Observatory, off the west coast of continental Africa (CVAO, $16°51'49"N, 24°52'02"W$). The observatory is located 50m from the coastline and is subject to persistent northeasterly trade winds. Meteorological conditions here have previously been well characterised by Carpenter et al. (2010), who observed a mean temperature of $23.6°C$, relative humidity of 79% and median wind speeds of

7.3 ms across a three year period. Given the placement of the observatory on the island, as well as the lack of major coastal features, this site is well suited for observations of unpolluted marine air.

Air was drawn vertically down a 10m stainless steel pipe with an inner diameter of 1" at a rate of 16.7L min $^{-1}$; from which an isokinetic inlet was used to draw a subsample of air at a rate of 0.3L min $^{-1}$. The sample line was heated to minimise condensation build up, and a Thermo PM10 inlet was used at the top of the pipe. Meteorological data was recorded atop a 30m

tower located at the observatory, with dominant wind direction recorded as one of sixteen compass points. This was funded as part of the Ice in Cloud Experiment - Dust (ICE-D) campaign, which had the aim of taking measurements of Saharan dust concentrations to better understand aerosol-cloud interactions and reduce uncertainty in numerical models. Details of the ICE-D surface experiments and accompanying aircraft sampling campaigns are described by Liu et al. (2018). The WIBS-4M was used to observe long-term trends in bioaerosol concentrations.

### 2.1.1 Intensive Campaign and Dust Samples

The monitoring component of the project was prefaced by a shorter term but highly intensive campaign, the results from which are used to inform the interpretation of the long term data. During the shorter campaign a WIBS-4A was used alongside a Laser Ablation Aerosol Time of Flight Mass Spectrometer (LAAP-ToF) (Marsden et al., 2018). Ground-based measurements using the LAAP-ToF and WIBS-4A were conducted over a 20 day period in August 2015 located near the Praia International

Airport, Cabo Verde ($14°570'N, 23°290'W$); 100m above sea level). A full description of the site and experimental setup is provided byLiu et al. (2018), and Marsden et al. (2019) but a brief summary is now given. Ground based ambient aerosol





measurements were made by the LAAP-ToF and WIBS-4 which were installed in the mobile Manchester Aerosol Laboratory, located 1500 m from the airport. Aerosols were sampled via a pumped inlet mounted on a 10 m tower. The total flow down the inlet was 1000L min$^{-1}$ and a series of aerosol instruments sub-sampled isokinetically downstream of a common manifold, which sub-sampled at 186L min$^{-1}$ from the main inlet. The inlet characterisation and losses are described in Liu et al. (2018)

and in the associated references. The LAAP-ToF measured the refractory aerosol in the size range 0.5-2.5$um$ whilst the WIBS-4 measured particles in the size range 0.5-20$um$, however the nominal cut-off of the inlet system was approximately 10$um$. A more detailed description of the WIBS and LAAP-ToF instruments is provided in section 2.2. The sampling tower was located upwind of the main airport terminal and the city of Praia (400 m and 2.5 km respectively) in the prevailing northeasterly winds so potential contamination sources were minimal. A full analysis anc classification of the back-trajectory wind fields during

this intensive experiment period was performed by Liu et al. (2018). They reported that dust plumes with Saharan origin were more frequently encountered during periods prior to 15 August, whereas dust from Sahara and sub-Saharan Africa influenced air masses were more frequently observed during mid July when winds were more easterly. They also noted that more efficient transport of dust via stronger easterly winds led to larger advected dust loadings with shorter dust transport times. These different dust advection pathways were reflected in different size and compositions of mineral dust recorded by the LAAP-ToF

during the intensive experiment (Marsden et al., 2016).

Calibration of the LAAP-ToF was performed with pure hematite samples (Liu et al., 2018), whilst the WIBS-4A was calibrated using NIST latex calibration beads and fluorescent glass beads, e.g. Crawford et al. (2015). In addition, subsequent to the ambient experiments Moroccan dust samples, hereby referred to as Morrocan Dust, were provided that were collected in

the Mhamid region ($29°51'43"N, 6°09'24"W$), which is thought to be representative of the dust material often observed in the continental outflow sampled here. These samples were dispersed through the WIBS-4M in a controlled laboratory experiment alongside additional collected samples of Illite. Sub-sets of these samples were gamma-irradiated to remove potential storage and transport contaminants. The UV-LIF response of the pure and irradiated samples were inter-compared and are described in the results section. Their respective fluorescent properties, as well as size and shape factors were also compared to the intensive

and long-term ambient observations. The negative ion spectra of the pure samples were also analysed using the LAAP-ToF in the laboratory for comparison. These sample properties were used to help contextualise the campaign results and the classification scheme used to identify bio-silicate particle concentrations.

## 2.2   Instrumentation

### 2.2.1   WIBS

Recent developments in UV-LIF technology have allowed real-time detection of bioaerosols, whereby spectral patterns correlate with specific particle characteristics (Huffman et al., 2019). This technology utilises the intrinsic fluorescent properties found within organic molecules, including proteins, co-enzymes, cell wall compounds and certain pigments to differentiate





between biological and non-biological particles. Each organic molecule has its own fluorescent emissions that are dependent on the incoming wavelength of the light that excites it. As such, it is possible for UV-LIF technology to discriminate between organic molecules and help to identify the types of aerosols being observed.

The WIBS-4M is a three channel UV-LIF spectrometer that uses Xenon flash lamps to excite particles with either 280nm

(Xe1) or 370nm (Xe2) light. The resulting emissions from excited fluorophores are then recorded by one of the instrument's two detection channels (De1 & De2), creating a 2x2 matrix. However, with the Xe1/De1 channel becoming supersaturated from elastically scattered UV light, only the other three resulting channels are considered. These are called Fl 1, 2 and 3, and are the product of Xe1/De2, Xe2/De1 and Xe2/De2 respectively. De1 is sensitive to 310-400nm UV light, while De2 covers a broader range from 420-650nm.

Common fluorophores considered in particle analysis include the amino acid Tryptophan, the co-enzyme NAD(P)H and the vitamin Riboflavin. Each fluorophore excites at 280nm, 270-400nm and 450nm respectively; and fluoresces from 300-400m, 400-600 and 520-565nm (Hill et al., 2009), (Lakowicz, 2013). The specificity of a biological compound's fluorescent properties enables particles to be classified into one of seven types, depending on whether they exhibit fluorescence in one, two, or three channels (Perring et al., 2015). Particle shape and symmetry is also measured through the use of a 635nm diode laser

that is initially involved in triggering the Xenon flash lamps to fire. The forward scattering light from a particle that passes through the laser hits a quadrant scattering detector. By applying a Mie scattering model, the distribution of the light on each quadrant can be used to estimate diameter, as well as an Asymmetry Factor between 0 and 100 (Gabey et al., 2010). This has been discussed in detail by Kaye et al. (2007), with values of 0 representing perfectly spherical particles and higher asymmetry factors describing more fibrous and linear shapes. When applying this range in the context of our observations, AF values of

10-20 reflect fairly spherical particles, with values of 30 or more beginning to represent ellipsoidal shapes.

### 2.2.2   LAAP-ToF

In single particle mass spectrometry (SPMS), an online measurement of aerodynamic size and composition is obtained with high temporal resolution. Several instrument designs have been described (Murphy, 2007), but a typical instrument features an aerodynamic lens inlet for particle beam creation and a high powered UV pulsed laser for laser desorption ionisation (LDI)

of single particles arriving in the ion source. Particle composition is analysed by time-of-flight mass spectrometry (TOFMS). Optical particle detection for temporal alignment of the UV laser pulse with the arrival of a particle in the LDI source region, and the determination of aerodynamic particle diameter by laser velocimetry.

The LAAP-ToF instrument (AeroMegt GmbH) used for this study featured a PM 2.5 aerodynamic lens, continuous wave (CW) fibre coupled optical detection lasers at 532nm, and LDI was performed by ArF excimer laser (model EX5, GAM Laser Inc.),

set to deliver 3–5 mJ per pulse of 193nm radiation. Bipolar TOFMS (TofWerks AG) was implemented for compositional analysis of positive and negative ions.

As a technique, SMPS is considered qualitative or semi-quantitative due to a strong matrix effect that occurs during the LDI of single particles. The ablation and ionisation process is incomplete so that competitive ionisation and charge transfer in the vapour plume results in a strong matrix effect (Reinard and Johnston, 2008). In addition, the reported particle number concen-





tration is affected by optical particle detection efficiency and the frequency of particles 'missed' by the pulsed UV laser. A full description of how these issues affected the instrument performance in this study can be found in Marsden et al. (2016), which estimates that the instrument was obtaining composition for 1% of dust particles in the size range $0.5 - 2.5um$.

Despite the limitations in quantitative analysis, the SPMS has proven useful for the online identification of particle composition

types, internal mixing state and their temporal trends in terms of particle number concentrations (Pratt and Prather, 2012). This typically proceeds using cluster analysis to differentiate particles classes and peak marker analysis to track changes in internal mixing state. In recent work, Shen et al. (2018) demonstrated the capability of the LAAP-ToF instrument to differentiate particle composition in a continental environment by a combination of cluster analysis and peak marker analysis.

## 2.3    Data analysis methods

A 9 sigma threshold was applied when setting fluorescent baselines for each WIBS model to remove weakly fluorescent inorganic particles that could otherwise act as interferences. This is a substantial increase in the threshold used from previously published work, but has been shown to significantly reduce the interference from mineral dust while maintaining the concentra-

tion of biological content. (Savage et al., 2017). As such, all fluorescent particles observed should contain significant amounts of biological material, and are subsequently considered to be a bioaerosol. It should be noted that these are not likely to all be single particles, but could instead be agglomerates, including biological particles attached to non-biological ones. The $D_{P50}$ of the WIBS-4M is $0.8um$. As such, only particles between 0.8 and $10um$ have been included in much of the analysis.

Agglomerative hierarchical cluster analysis was performed using Ward linkage with Euclidean distance for the fluorescent

particles, using the method described in Crawford et al. (2015). This has been successfully tested in the past when applying a controlled set of fluorescent polystyrene latex spheres (PSLs) through the same instrument model (Crawford et al., 2015), (Robinson et al., 2013), and has also been shown to accurately capture bioaerosol mixtures within a semi-arid forest (Gosselin et al., 2016). This process begins with individual data points being treated as their own cluster, that are sequentially combined into larger clusters based on the 'distance' between data point values. Ward linkage identifies which pair of clusters would yield

the minimum increase in total within-cluster variance after merging, and consequently merges them next. A Calinksi-Harbasz score is then calculated, suggesting an optimum number of clusters.

The transmission efficiency of the inlet as a function of size has also been accounted when considering size distributions for particles up to $10um$. Inlet penetration as a function of size has also been considered, based on the Andersen 321A PM10 inlet (McFarland, 1984).


Analysis of the LAAP-ToF data acquired during the ICE-D intensive campaign has been presented by Marsden et al. (2019), in which the mineralogy and mixing state of North African dust transported to the Cape Verde Islands is described in detail with reference to dust samples collected on the ground from potential source areas in North Africa. Using a combination of cluster analysis and sub-compositional analysis, that work showed that differences in mineralogical composition of ground



samples can be detected on a regional scale, whilst variation in internal mixing state are much more localised. In transported dust however, the mixing state was much less varied suggesting that local sources become well mixed during transport.

In the present study, we used the compositional classes already identified in the above work by cluster analysis, and compared the temporal trends in particle number concentrations with that of the biological particles detected by the WIBS. In addition,

5    we used mass spectra of the previously identified mineral dust class to examine internal mixing of dust with biological material using peak marker analysis. More specifically, biological markers CN- (m/z 26) and CNO- (m/z 42) were used to identify those mineral dust particles which had a significant marker of internal mixing with biological material, and also compared the temporal trends in those particle number concentrations to the biological particles detected by WIBS.

This approach is similar to that used by Zawadowicz et al. (2019), who examined the biological content of dust particles in

10   North America by comparing temporal trends in particle number of a WIBS with SPMS, in that case the PALMS instrument. However, in our study we have used different criteria for biomarkers in the mass spectra, using chlorine to normalise the biomarker signals rather than phosphorous. Chlorine is a ubiquitous signal in transported mineral dust, and is easily ionised because of its high electro affinity. This reduces the temporal incoherence in biomarker ion signals caused by the matrix effect.





# 3    Results

## 3.1    Long-Term Observations

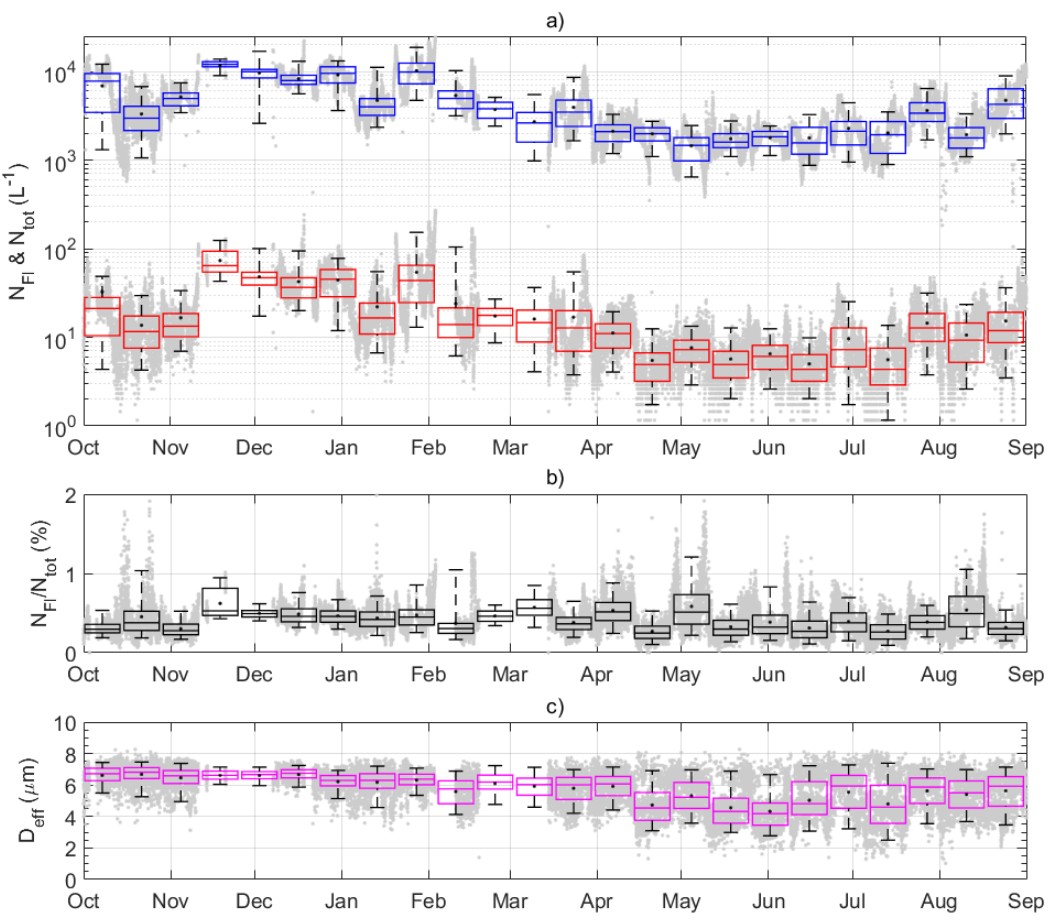

**Figure 1.** Panel a) represents a Box and whisker plot of both average fluorescent (red) and total (blue) particle concentrations from $0.8\text{-}10um$. Whiskers represent the 5th and 95th percentile values. Boxes represent 14 day average values, with the mean shown as a black dot. Grey dots indicate 15 minute data. Panel b) shows the ratio of fluorescent particle concentrations relative to total particle observations, while panel c) shows the trend for the Effective Diameter of fluorescent particles.

A clear seasonal trend for fluorescent and total particle concentrations can be seen in Fig.1a, with both being highest during winter and lowest in summer. Monthly median bioaerosol concentrations as high as $45L^{-1}$ were found with 95th percentile values exceeding $130L^{-1}$ during strong dust events. Such peaks in concentrations are short-lived and regular. Fig. 1b suggests





that bioaerosols are a minority particle type, with non-fluorescent concentrations consistently dominating particle contributions. The fraction of fluorescent particles compared to total particle concentrations was a mean value of $0.4 \pm 0.2\%$. The $99th$ percentile value was a fraction of $1.1\%$. Some high fraction events exceeding $1.5\%$ can be seen during the campaign, for example in Oct and May. These will be discussed in greater detail further on. Fig. 1c highlights how larger particles are observed

5   during winter. Note that there is no relationship between the total particle concentration and fraction of fluorescent particles ($r^2 = 0.02$), but a strong correlation exists between fluorescent and non-fluorescent particle concentrations ($r^2 = 0.79$).





## 3.2 Cluster Products

Agglomerative hierarchical cluster analysis was performed using Ward linkage with Euclidean distance for all fluorescent particles. A four cluster solution produced the greatest Calinski-Harbasz score, with information regarding their fluorescence in each channel, size, and asymmetry shown in Table 1.

| | Fl 1 [a.u.] | | | |
| --- | --- | --- | --- | --- |
| | 0-5um | 5-10um | 10-20um | All |
| Cluster 1 | 1219.7± 497.0 | 1033.1± 372.8 | 1114.4± 361.6 | 1169.3± 469.3 |
| Cluster 2 | 25.2± 64.1 | 34.5± 70.7 | 34.9± 85.7 | 27.9± 66.5 |
| Cluster 3 | 8.0± 26.1 | 49.9± 63.7 | 69.8± 0 | 8.0± 26.1 |
| Cluster 4 | 15.7± 52.4 | 64.7± 110.9 | 79.2± 133.5 | 25.4± 72.2 |
| | Fl 2 [a.u.] | | | |
| | 0-5um | 5-10um | 10-20um | All |
| Cluster 1 | 22.9± 82.7 | 132.9± 221.9 | 453.8± 453.3 | 80.1± 215.5 |
| Cluster 2 | 24.0± 44.1 | 43.9± 61.1 | 68.7± 72.9 | 30.3± 51.0 |
| Cluster 3 | 39.7± 57.0 | 27.0± 36.0 | 64.3± 0 | 39.7± 57.0 |
| Cluster 4 | 309.0± 299.0 | 495.1± 276.1 | 618.2± 373.1 | 348.5± 313.9 |
| | Fl 3 [a.u.] | | | |
| | 0-5um | 5-10um | 10-20um | All |
| Cluster 1 | 10.5± 56.0 | 99.5± 231.2 | 366.8± 413.8 | 57.8± 195.1 |
| Cluster 2 | 16.1± 48.9 | 22.6± 51.0 | 47.2± 70.9 | 18.5± 50.3 |
| Cluster 3 | 25.3± 54.4 | 23.5± 64.7 | - | 25.3± 54.4 |
| Cluster 4 | 247.5± 319.7 | 478.4± 314.7 | 611.5± 407.3 | 296.0± 342.2 |
| | Size | AF | % | # |
| Cluster 1 | 4.9± 3.2 | 19.7± 16.3 | 0.4± 1.3 | 7297 |
| Cluster 2 | 4.4± 2.1 | 23.4± 14.8 | 58.8± 17.7 | 893,241 |
| Cluster 3 | 1.6± 0.5 | 7.0± 4.9 | 34.7± 15.9 | 547,711 |
| Cluster 4 | 3.4± 3.4 | 16.5± 16.2 | 6.2± 5.5 | 121,455 |

**Table 1.** Characteristics of individual clusters when using a four cluster solution for exclusively fluorescent particles. ± represents one standard deviation, calculated using raw, non-zero particle data. AF represents Asymmetry Factor as given by the WIBS-4M and % reflects the contribution each cluster makes to the total fluorescent particle observations. The column # represents the number of particles classified within a given cluster.



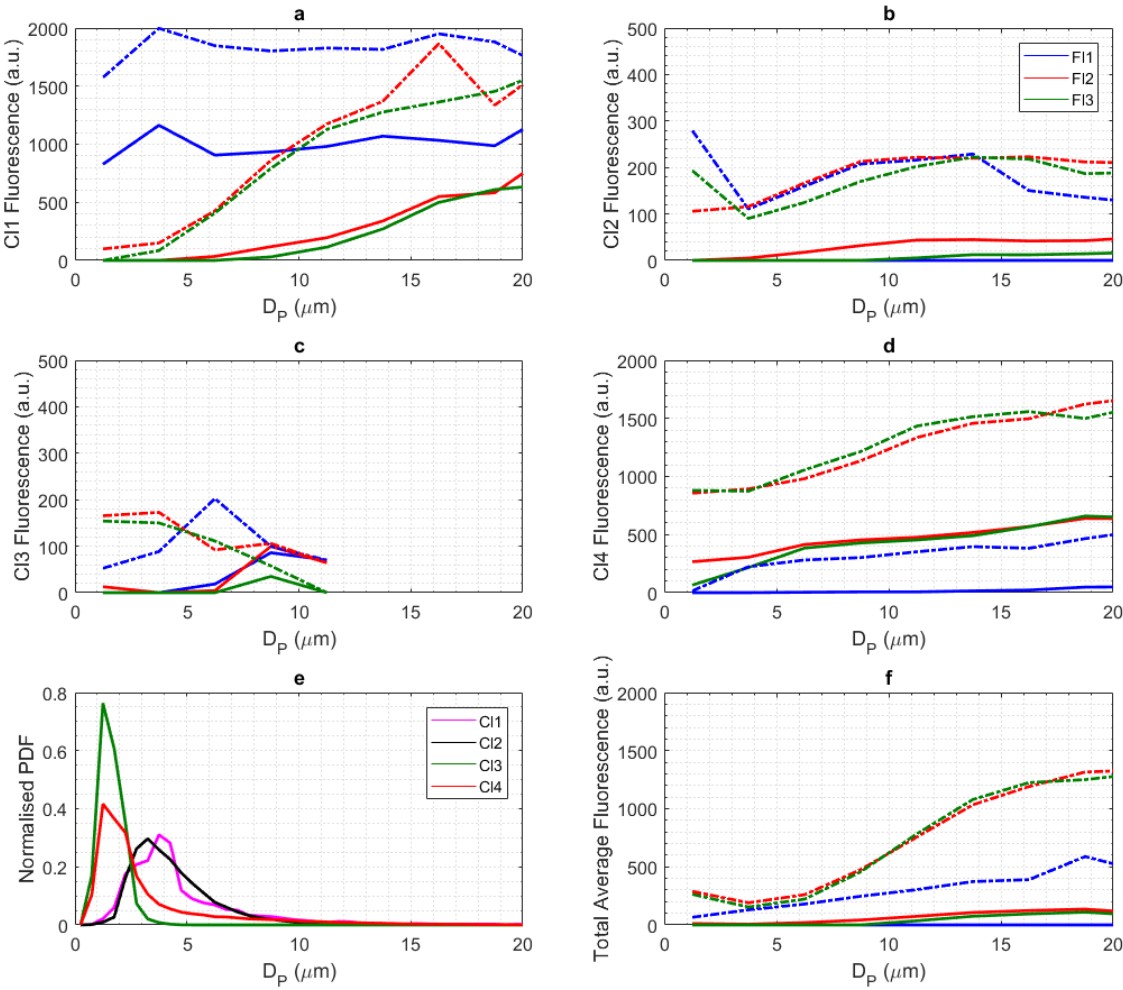

**Figure 2.** Solid lines represent the median fluorescence of particles as a function of size, instrument channel and cluster. Dotted lines represent 95th percentile values. $2.5um$ bins have been used for all fluorescence traces. Panel e. represents the normalised size distribution of fluorescent particles across all clusters, using $0.5um$ bins.

All fluorescent particles seen across the long-term campaign were included in Fig.2. Fluorescence generally increases with size, but this trend is not equal across all channels and clusters. It is strongest for Clusters 1 and 4, which are also more fluorescent across most size ranges. Panel e) shows the small size of observed particles, with Clusters 3 and 4 being much smaller than Clusters 1 and 2. Panel f) shows the median fluorescence in each channel across all fluorescent particles combined.





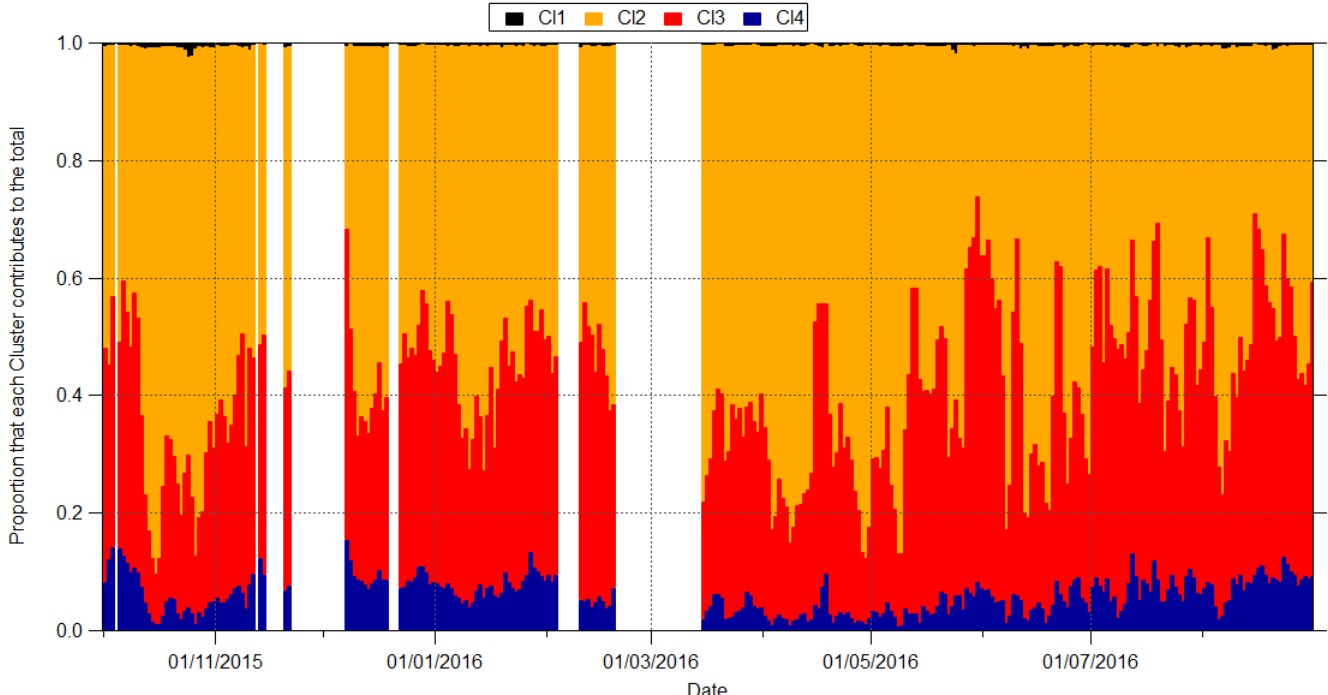

**Figure 3.** The proportion of the total fluorescent material that each cluster contributes over the 11 month monitoring campaign. Values are taken using 1-day integrations of cluster concentrations between 0.8 and $10um$. Any gaps reflect missing data.

The relative proportion that each cluster contributes to the total fluorescent concentration shows some variability, but no clear trend. This would suggest that Cape Verde is generally exposed to the same particles year-round. Clusters 2 and 3 are clearly the two dominant clusters, accounting for a combined total of approximately $90\%$ of fluorescent particles. When compared to the average fluorescence values in Fig. 2, these two clusters notably have the weakest fluorescence.





## 3.3 Back Trajectory Analysis

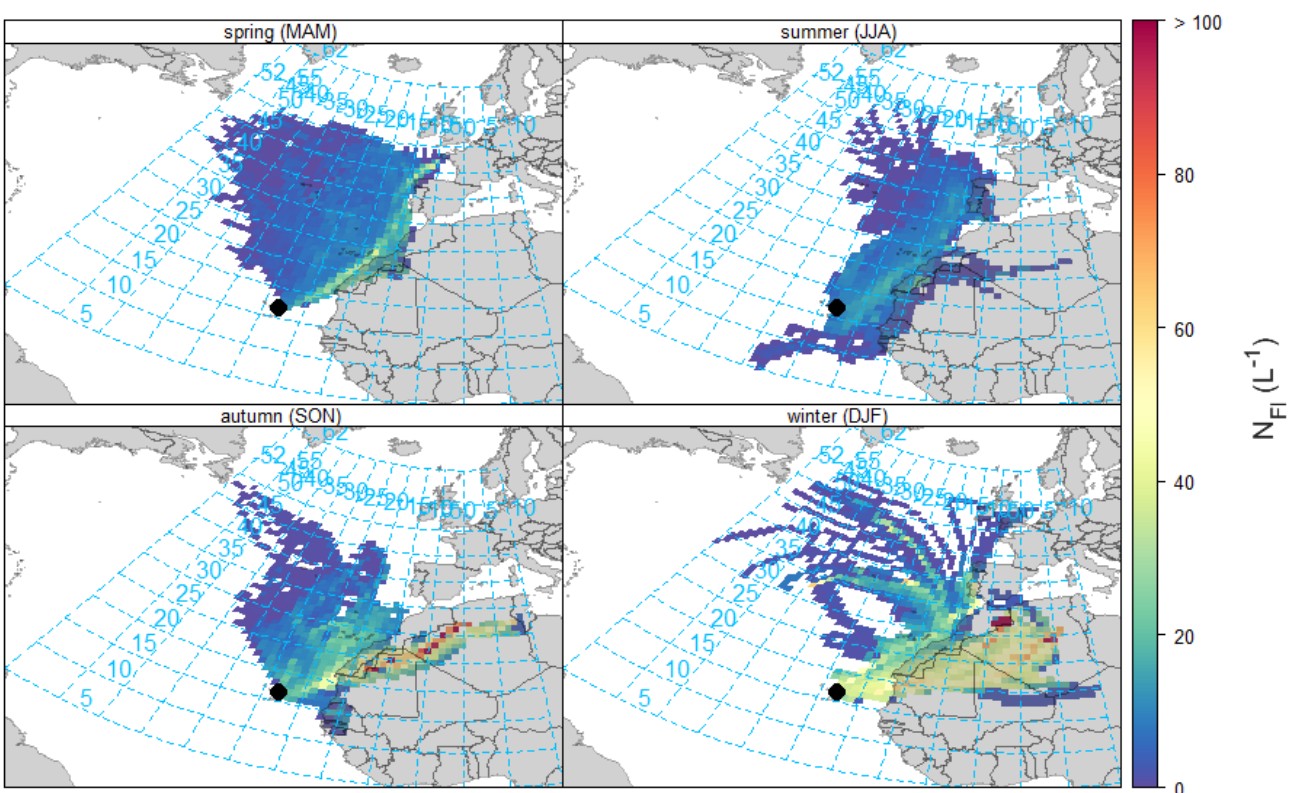

**Figure 4.** 120 Hour back trajectory analysis for fluorescent particle concentrations using Hysplit integrated with the Openair package. Note that autumn is using 2015 data, as is the month of December during winter. Spring and summer are using 2016 data. The colour scale has been capped at $100\ L^{-1}$ for visual purposes and only particles between 0.8 and $10um$ have been included.

120 Hour back trajectories using NOAA's Hysplit merged with the Openair package (Car) were calculated from the observatory, with a starting height set at 10m. Fig. 4 captures the enhanced fluorescent particle concentrations seen during periods of continental outflow as compared to Atlantic or coastal back trajectories. Furthermore, it appears that continental trajectories predominantly occur during the same months that the increase in fluorescent and total particles are seen (Oct-Feb). Back trajectories were also calculated for periods where the fluorescent fraction was greatest. Particularly high ratio events can be identified during Oct, May and Aug in Fig. 1b. These events all show coastal back trajectories, suggesting coastal sites have higher relative biological content.

Fluorescent particle properties were also investigated as a function of trajectory direction. Three weeklong periods were chosen, where the trajectories were consistently coming from a specific region. These regions were oceanic, coastal and conti-





nental. Oceanic was defined as trajectories that spent the previous 120 hours passing across the Atlantic Ocean, never passing landmass. Coastal was defined as northeasterly trajectories passing along the edge of the African continent and continental was described as more easterly trajectories passing through the Sahara. As it was important to have an entire week of trajectories coming from one source, each of these regions were picked out at different times of the year. The continental period took place

5 25/01/2016 - 31/01/2016, the coastal from 20/06/2016 - 27/06/2016 and the marine from 20/04/2016 - 26/04/2016. Average fluorescence across the 4 clusters and 3 channels have been compared for particles from each region, as shown in Fig.5. It does not appear that there are significant differences in the fluorescent properties of particles depending on their source region.

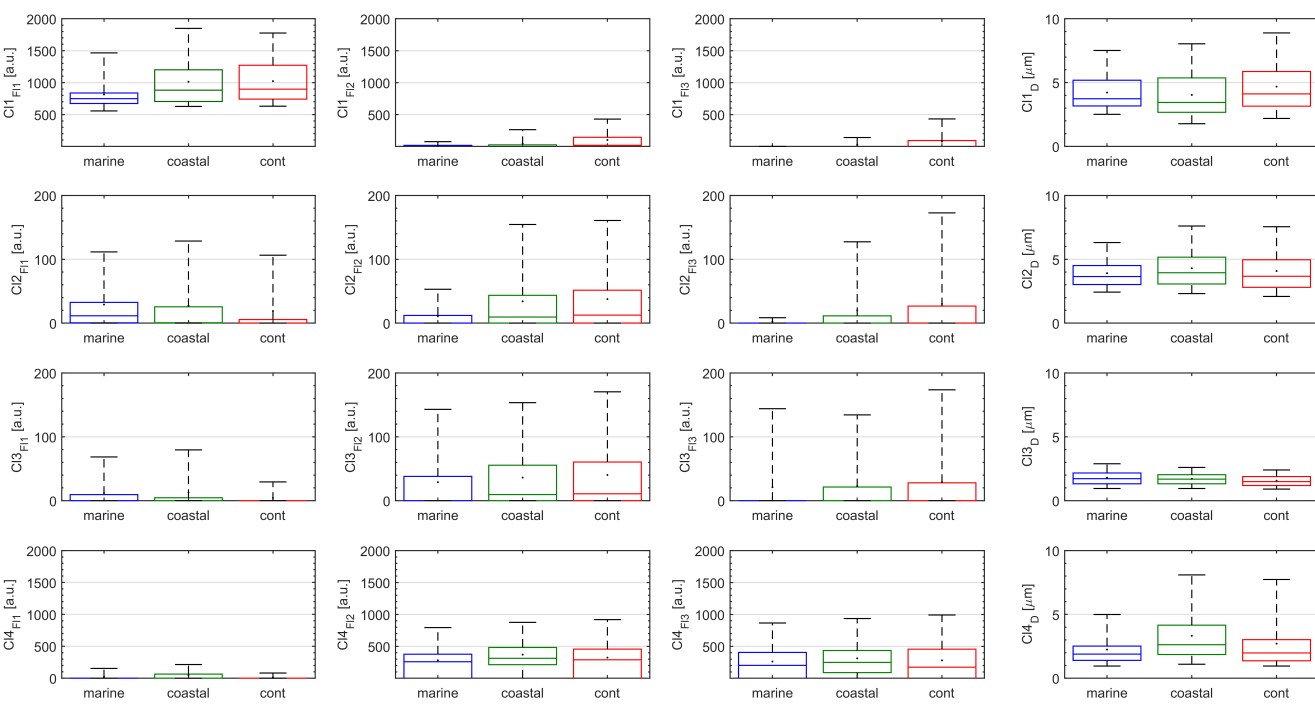

**Figure 5.** Mean fluorescence and size properties for each cluster, depending on the pathway of the previous 120 hour trajectory. Each region (marine, coastal and continental) have taken one week's worth of data, from 20/04/2016 - 26/04/2016, 20/06/2016 - 27/06/2016 and 25/01/2016 - 31/01/2016 respectively. Columns represent Fl channels 1-3 with the final column representing size in $um$.





## 3.4 Comparison to Laboratory Samples

All of the dust samples dispersed through the WIBS-4M show relatively weak fluorescence across all three channels, but are on average more fluorescent than clusters 2 and 3. This is particularly true for Moroccan dust in Fl 1. Exposing these samples to gamma irradiation produced little effect, marginally reducing the fluorescent properties of Moroccan dust while increasing the fluorescence of the Illite sample. The size and asymmetry values are quite similar across all samples, and compare most closely with those of Cluster 2.

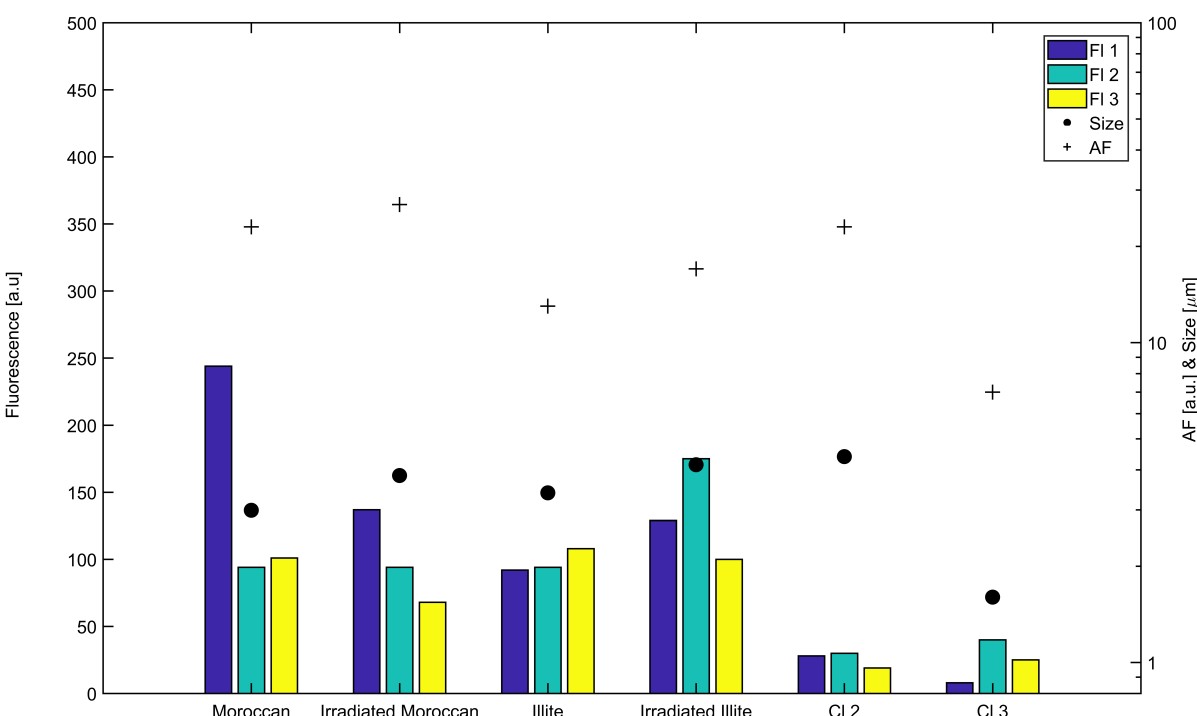

**Figure 6.** Mean fluorescence, size and shape properties of the 4 dust samples put through the WIBS-4M in a laboratory environment and their comparison to the two dominant long-term campaign clusters - Cluster 2 and 3.



## 3.5 Comparison to ICE-D LAAP-ToF Measurements

Analysis of the negative ion spectra revealed that sea-spray aerosol was the dominant aerosol detected at the site by the LAAP-ToF, accounting for approximately 87% by number. The remainder was comprised of silicate mineral dust ( 5%), calcium chloride ( 3%) and secondary material ( 4%). Cluster analysis of the WIBS-4A data produced a 4 cluster solution which was

5    similar to the long term CVAO cluster solution, featuring two weakly fluorescent dominant clusters and two highly fluorescent, likely Primary Biological Aerosol Particle (PBAP) clusters. A number concentration time series of select LAAP-ToF and WIBS-4A data products are shown in Fig. 7 , where similarities in the trends of silicate dust and WIBS-4A Cluster 3 can be observed during a dust event (10/08 onwards), suggesting that this cluster may represent a bio-dust mixture which will now be examined further.

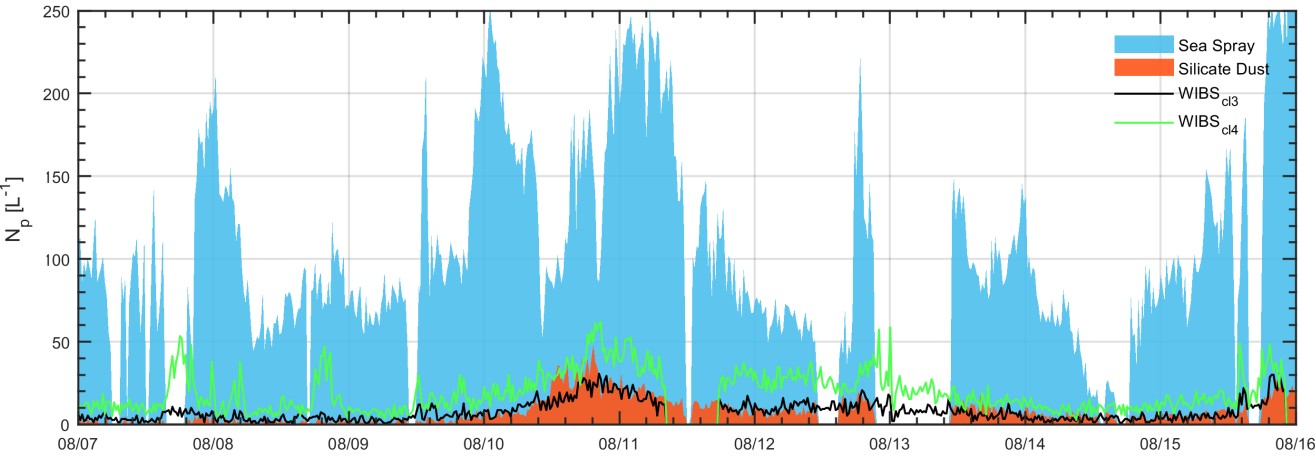

**Figure 7.** Time series of sea spray aerosol and silicate dust particle numbers concentrations determined by LAAP-ToF and WIBS-4A clusters Cl3 & Cl4. 20 Minute averages have been used here.



Deeper analysis of the LAAP-ToF silicate dust class was performed to screen for potential bio-markers. Fig. 8 shows example negative ion spectra from laboratory samples of bacteria (top panel) and c-means fuzzy cluster products from ICE-D representative of pure dust and dust containing bio-markers (middle and bottom panels respectively). It can be seen that the silicate bio spectrogram contains silicate m/z peaks (e.g., $SiO_2$- $SiO_3$-) and also bacterial bio-markers (e.g., $CN^-$  $CNO^-$)

5  which are not present in the pure dust spectra, suggesting that a subset of the observed dust is mixed with biological material. The LAAP-ToF silicate dust is then filtered for bio-markers using the ratio of bio markers to chlorine (CN+CNO/Cl ).

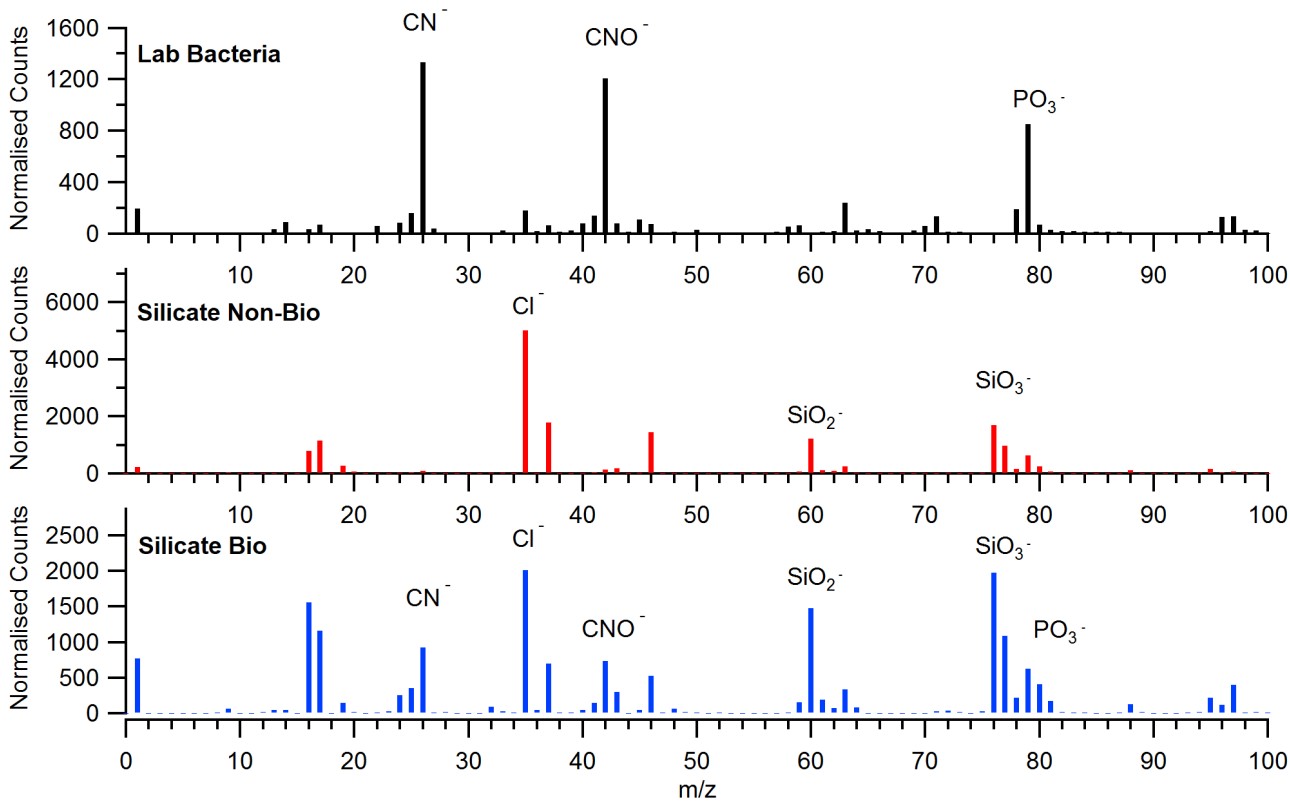

**Figure 8.** Average negative ion spectra of laboratory generated bacteria and the ICE-D silicate dust products.





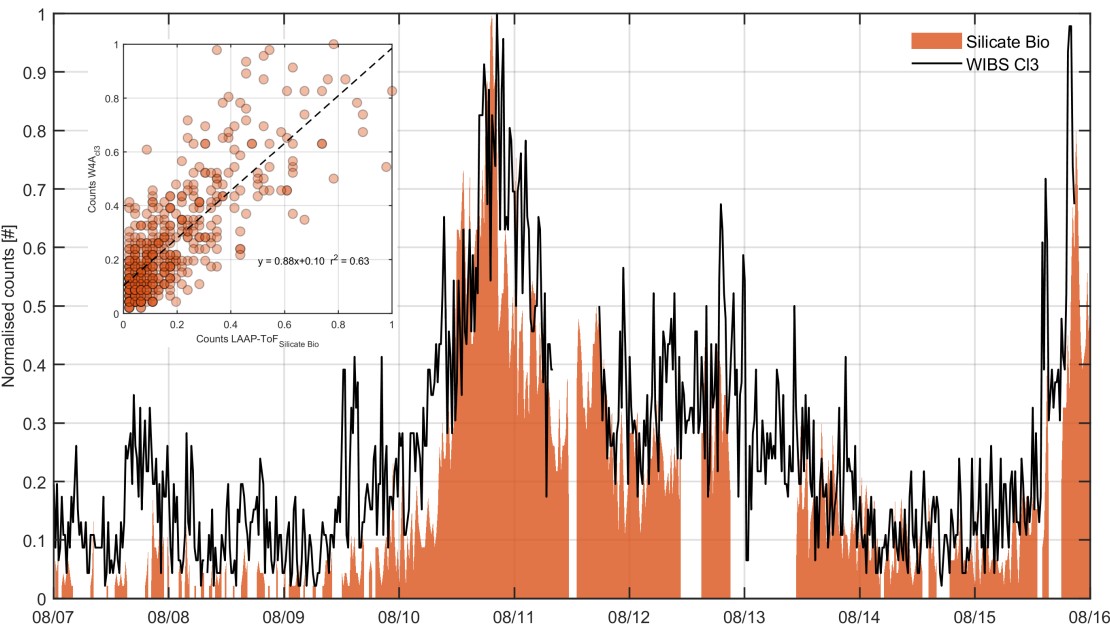

**Figure 9.** Time series of WIBS Cl3 and silicate dust filtered for Bio markers using 20 minute averaged number counts normalised by the maximum observed concentration for each instrument.

A time series of the LAAP-ToF silicate bio and WIBS-4A Cluster 3 products are shown in Fig. 9. Here it can be seen that the general trends are in good agreement ($r^2 = 0.63$). This suggests that the WIBS is sensitive to dust particles which are mixed with bacteria, and that these particles can be segregated from the general fluorescent aerosol population and classified using the cluster analysis method described in Section 2.3. There is also a moderate correlation between the WIBS-4A Cluster 3 and pure silicate ($r^2 = 0.49$) as expected, since the silicate bio will be a subset of the pure silicate. The correlation between Cluster 3 and sea spray is poor ($r^2 = 0.20$), affirming that our observations of fluorescent particles are genuine, and are not being skewed by known interferents.





## 4 Discussion

### 4.1 Long-Range Transport and Trade Winds

The identified peaks in particle concentration are consistent with back trajectory analysis showing the sources to be Saharan. Strong Aeolian generating mechanisms mean that potentially significant concentrations of soil bacteria could have attached themselves to dust particles, which are acting as an agent for their distribution and transport. More specifically, it is likely to be the Harmattan Wind that is driving these events. This is a north-easterly trade wind that blows across Africa from approximately November to March, although high concentrations of dust are already present almost all year round (Prospero, 1999). Source regions in Tunisia and northern Algeria have been identified before, as well as dust that has been transported from the Chad basin in the Bodele Depression (Herrmann et al., 1999). Such Harmattan dust clouds have often been found to affect the coastal regions of the Gulf of Guinea. These clouds are a major annual event, noted for exacerbating cardio-vascular health issues and increasing daily mortality by 8.4% (Perez et al., 2008). Previous studies by Enete et al. (2012) found this dust to contain high quantities of silicon, which is consistent with the LAAP-ToF results presented here.

It is interesting that continental back trajectories during summer do not contain similar concentrations of fluorescent material, despite a similar fluorescent ratio and wind speeds to those seen in winter. This is likely due to the seasonal shift in the Intertropical Convergence Zone (ITCZ), as outlined by Chiapello et al. (1995). During summer the ITCZ is located between $6 - 10°N$ and dust gets transported into the Saharan Air Layer (SAL) at a maximum altitude of 7000m. At this height there is little opportunity for the dust to mix with trade winds, and it is instead transported significant distances across the Atlantic. However, during winter the ITCZ shifts south and the dust is no longer able to get taken into the SAL. This constrains the dust into lower altitudes, where they are subject to mixing with trade winds, including the Harmattan Wind. Dust consequently gets deposited much sooner and often blankets Cape Verde. The ecological consequences of this have been investigated by Korte et al. (2017), who placed sediment traps across the Atlantic to measure fluxes in biogenic constituents, including biogenic silica. They observed higher biogenic fluxes during winter, owing to Saharan dust getting deposited during the previously discussed trade wind events. They also observed a seasonal maxima of biogenic silica towards the west, suggesting that the bioaerosols we observe are not as readily deposited into the ocean as some other particle types.

It is also interesting to consider potential differences in dust from the various source regions. This has been investigated by Patey et al. (2015), who also monitored dust concentrations at the CVAO. By looking at specific elemental ratios within dust samples, they were able to identify where the dust was thought to have come from, as well as what it was comprised of. They found during summer months that 92.5% of samples contained a contribution from Sahel, compared to just 52.3% of samples collected during winter. These observations are supported by previous work from Prospero and Lamb (2003), who found that it was Sahelian dust that was predominantly transported across the Atlantic via the SAL, during summer.





## 4.2    Identification of Fluorescent Material

The relative contributions of each cluster in Fig.3 highlight the importance of Clusters 2 and 3. Furthermore, it shows that the mixture of bioaerosols have no clear pattern, with each cluster present all year round. The intensive measurements from the LAAP-ToF and WIBS-4A have provided supporting evidence to suggest that most of the fluorescent particles observed are

mixtures of biological and non-biological material i.e. bacteria attached to dust. It is likely because of this that fluorescence intensity is correlated with particle size. Larger particles may carry greater numbers of bacteria due to increased surface area, and as a result should carry more fluorophores for detection by the instrument. The relationship between fluorescence and particle size has been investigated extensively by Hill et al. (2015), who have shown a generally positive correlation.

Concluding that Clusters 2 and 3 are at least partially bacterial in nature is in contrast to findings by Savage et al. (2017), whose laboratory tests on *Bacillus atrophaeus, Escherichia coli and Pseudomonas stutzeri* have produced different fluorescent spectra to those observed here. In their tests bacteria have been shown to fluoresce strongly in Fl 1 and weakly in Fl 2 and 3, while our results show little increase in Fl 1 concentrations during winter, but substantial increases in Fl 2 and 3. However, it must be emphasised that they used 'pure' bacterial samples, potentially unrepresentative of the mixed aggregates we observed.

Furthermore, they outline previous work by Agranovski et al. (2004), who found that a UV-APS was effective in identifying fluorophores at similar wavelengths that a WIBS-4A was unsuccessful in doing so. They hypothesise reasons for this, including potential differences in gain voltages applied in the instrument, and weaker excitation intensity in Xe2 with respect to Xe1. With that considered, it must also be noted that there are high levels of agreement between the WIBS-4A used in the intensive measurements and the WIBS-4M for 11 months. As such the more likely explanation may simply be differences between

the samples used in the laboratory and those measured at the observatory. Without proper characterisation of a bacteria/dust aggregate in a controlled setting it is difficult to interpret how this could impact the emissions spectra. Internal re-absorption of fluorescence is known to occur when emitted fluorescent wavelengths overlap with a particle's absorption spectrum. In this instance it is possible that the dust absorbs certain wavelengths emitted from the bacteria, skewing what is detected by the instrument. It must be emphasised that applying a 9 sigma threshold to the fluorescent baseline of the WIBS-4M means that

our observations are unlikely to be interferents.

Clusters 1 and 4 are interesting because of their high fluorescence intensity. It is possible that these clusters represent 'pure' bioaerosols, that have not mixed with non-fluorescent material or are relatively more exposed due to for example larger bacterial aggregates present. Cluster 1's spectral profile matches closely with laboratory experiments using bacteria (Savage et al.,

2017), with a strong fluorescence in Fl 1 seen across all size ranges. However, Cluster 4 remains difficult to identify. It shows similarly high intensity fluorescence to Cluster 1, but only in Fl 2 and 3. It's relative contribution to the total bioaerosol count is significant, yet given the location of the observatory seems unlikely to be other common terrestrial bioaerosols such as pollen fragments. Although it may therefore seem likely to have a marine source, it's concentrations follow similar seasonal trends to the other clusters which are more easily explained by Aeolian events. With potential differences in their capacity to act as ice





nucleators when compared to bacteria and dust aggregates, it is useful to quantify their concentrations.

The dust samples dispersed into the WIBS-4M also support our argument that bacteria/dust aggregates are our dominant bioaerosol type. Many of the sampled particles shared similar spectral profiles to those for Clusters 2 & 3, being weakly fluorescent in all three channels. Each sample also predominantly consisted of non-fluorescent particles, sharing a similarly low fluorescent particle ratio to our campaign observations.

### 4.3 Comparison to Previous Studies

The results presented here are broadly consistent with previous studies demonstrating that mineral dust is often observed to be mixed with biological material e.g., Yamaguchi et al. (2012), Griffin et al. (2001) and Maki et al. (2018). Similar to the approach used here, a recent study by Zawadowicz et al. (2019) assessed the prevalence of biological material over the continental United States using single particle mass spectrometry and UV-LIF, finding that 30 to 80 % of biological particles were mixed with mineral dust. While they provide evidence for bio-mineral mixtures in their study, the fluorescent number concentration derived from the WIBS is likely an overestimate of the true bioaerosol concentration due to the choice of 3 sigma thresholding including pure mineral dust interferents in their assumed bioaerosol population, as a small, but significant, subset of mineral dust naturally exhibits weak fluorescence (Huffman et al., 2019), (Savage et al., 2017), (Crawford et al., 2016), (Pöhlker et al., 2012). The conservative 9 sigma threshold used in our study excludes these non-biological interferents from the presented PBAP classes (Savage et al., 2017). Furthermore, the use of only negative ion spectra makes resolving bio-minerals from pure minerals challenging. The use of both positive and negative ion spectra in our study provides greater particle information and consequently improves our ability to classify bio- and pure minerals as distinct groups (Shen et al., 2018).

### 4.4 Summary and Conclusions

This study has utilised UV-LIF technology to provide long-term measurements of bioaerosol concentrations within an important but often overlooked region of the world. Seasonal variation in both fluorescent and total particle concentrations are clearly observed, likely as a result of the annual patterns of the ITCZ and subsequent mixing with trade winds. This can be readily seen from the Hysplit back trajectories, with the highest particle concentrations coming from mainland Africa during winter months.

When considering the source regions in the Sahara and significant correlation between fluorescent and non-fluorescent particle concentrations; it is presumed we are observing high mineral dust concentrations with some associated bacteria. This is supported through the LAAP-ToF and WIBS-4A intensive measurements; with a significant correlation between the LAAP-ToF's silicate bio counts and the concentrations of a clustered subset from the WIBS-4A.

Cluster analysis results from the WIBS-4A compare favourably with those from the WIBS-4M, with both suggesting a four cluster solution that share similar fluorescent profiles. For the long-term campaign Clusters 2 and 3 dominate fluorescent par-



ticle contributions, accounting for approximately 90% of all bioaerosols. Both are weakly fluorescent, but with a 9 sigma threshold having been applied are unlikely to be interferents.

A laboratory experiment using representative dust samples has shown similar fluorescent properties to these clusters, helping to contextualise our observations. These presumed bacteria and dust aggregates are still a minority particle type, accounting for a mean 0.4±0.2% of total coarse aerosol concentrations. Although this ratio is relatively low, it should be noted that the raw number of bioaerosols present is still quite high, with monthly median concentrations as high as $45L^{-1}$ and 95th percentile values exceeding $130L^{-1}$.

Highly fluorescent and likely primary bioaerosols have also been identified in Clusters 1 and 4, accounting for an average 6.6% of total fluorescent particles. These have not be conclusively identified, but it should be stated that Cluster 1 most closely resembles the spectral profile of pure bacteria outlined by Savage et al. (2017), while Cluster 4 remains unidentified.

Our long-term measurements are consistent with the observations of Korte et al. (2017), who made an association between the deposition of biogenic silica along the Atlantic and high levels of dust from the African continent. It would be interesting for future work to determine whether there are microbial differences within this dust when compared to other regions, following the approach described by Maki et al. (2018). Either the presence of more efficiently ice nucleating bacteria strains or simply greater concentrations could potentially explain why Sassen et al. (2003) found dust in African outflow to ice nucleate at significantly warmer temperatures than similar studies by Ansmann et al. (2008) and Zhang et al. (2012). Such work would have the capacity to improve Global Climate Model (GCM) simulations.





## Appendix A: Appendix

**Figure A1.** Summary of the average fluorescence, size and AF of each cluster from the long-term campaign.

*Competing interests.* There are no known competing interests present in the execution and publication of this project.



*Author contributions.* D.Morrison is a PhD student and primary author for this paper, responsible for most written components. I.Crawford processed the data, contributed to the analysis and provided guidance on the paper's contents. N.Marsden was involved in operating and analysing the data from the LAAP-ToF and has collaborated with I.Crawford to write the sections pertaining to the LAAP-ToF. He has also sourced the dust samples for the lab experiments. M.Flynn was involved in the planning and execution of both the short-term and long-term components of the project, as well as providing an estimate of the transmission efficiency of the sampling line. K.Read manages the CVAO WMO-GAW station and provided support and access to facilities for this experiment as part of ICE-D, including meteorological data which is archived at the WMO-GAW and British Atmospheric Data Centre (BADC). N.Luis was responsible for instrument operation and maintenance during the long-term campaign. V. Foot is part of DSTL and provided technical support and loan of UV-LIF instrumentation. P.Kaye and W.Stanley helped in maintenance and repair of the instrument. H.Coe was involved in the short-term component of the project, while D.Topping has provided guidance on the direction and written components of the paper. M.Gallagher has overseen the entire project, acting as the primary lead. He is the supervisor of D.Morrison, and has offered guidance at every stage.

*Acknowledgements.* This project was funded by NERC as part of the ICE-D campaign (NE/M001954/1). D.Morrison's PhD studentship was funded by the NERC Doctoral Training Program (DTP). I.Crawford has been funded as part of the BIOARC campaign (NE/S002049/1). We thank DMT for the loan of the WIBS-4A used in the LAAP-ToF comparison.

*Data availability.* Due to the large file sizes for the dataset, it is available upon request to the lead author.



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
