# Peer review of "Quantifying Bioaerosol Concentrations in Dust Clouds through Online UV-LIF and Mass Spectrometry Measurements at the Cape Verde Atmospheric Observatory"

_Atmospheric Chemistry and Physics, 2020_

## Referee Comment (RC1) · Anonymous Referee #1 · 27 Apr 2020

Morrison et al. submitted a manuscript titled "Quantifying Bioaerosol Concentrations in Dust Clouds through Online UV-LIF and Mass Spectrometry Measurements at the Cape Verde Atmospheric Observatory." This manuscript presents annual observations of bioaerosol concentrations associated with long-range dust transport. UV-LIF single-particle measurements were compared with LAAP-ToF measurements, and the results show high correlations between LAAP+ToF spectral signatures indicating particles mixed with bio-silicate mass and UV-LIF fluorescent particle signatures. The manuscript is well written, and the experimental/analysis methods are clear as well as

the results. I support the publication of this manuscript with some minor technical edits. Here are a couple of suggestions for technical edits:

-Line 9: "A full analysis anc classification of the back-trajectory wind fields during this intensive experiment period was performed by Liu et al. (2018)." The word "anc" should be corrected to "and".

-Line 30: A full description of the site and experimental setup is provided byLiu et al. (2018), and Marsden et al. (2019) but a brief summary is now given. Suggestion to put a space between "byLiu"

---

## Referee Comment (RC2) · Anonymous Referee #2 · 2 Jul 2020

*Review for manuscript acp-2020-157, "Quantifying Bioaerosol Concentrations in Dust Clouds through Online UV-LIF and Mass Spectrometry Measurements at the Cape Verde Atmospheric Observatory."*

In this study, the authors describe long-term observations of fluorescent particles at Cabo Verde Observatory collected using a UV-LIF instrument, the WIBS-4M. They also use observations of fluorescent bioparticles and dust particles (collected by a single particle mass spectrometer, a LAAP-ToF) from the ICE-D intensive field campaign to inform the interpretation of the longer WIBS dataset in regard to mixed dust-bioparticle types. They found that ~0.5% of all particles measured by the WIBS contained biological material, and that particle concentrations were elevated due to terrestrial influences. They also report a WIBS cluster that correlates with mixed-dust particles from the ICE-D field campaign.

There is considerable interest in both measuring bioaerosol contributions and distinguishing between dust, bioparticles, and mixed dust bioparticles. There are few long-term observational studies, so the topic is worth study. My recommendation is that the manuscript is not published without major revisions.

General comments

Use of a forced trigger+9 σ threshold for WIBS measurements has been shown to reduce potential non-biological interferents mineral dust samples, but not for all particle types, e.g. soot. This is particularly important because much of the analysis in the paper depends upon the assumption that all fluorescent particles are biological (page 7, line 15). This should be addressed somehow.

Using two different versions of the WIBS is a weakness of the study. I think that some context on the difference between the 4A and 4M would be helpful in order to understand to how equivalent these two datasets are. Additionally, I found it difficult to tell whether the particle clustering from the two WIBS datasets was performed together or separately. If so, some comparison between the two sets of clusters would be helpful to make sure that they are really equivalent. Also, how was the WIBS-4M calibrated? The manuscript only mentions the 4A.

More information is needed on the back-trajectory analysis. How many were released, and how often? Some context on what the Openair package analysis provides would be helpful for the reader.

For the LAAP-ToF measurements, it should be noted that simply using $CN^-$ and $CNO^-$ as boolean-type markers for biological material is suspect. They are pretty non-specific and found in a number of different particle types. This is discussed in detail by Zawadowicz et al. (2017). While these markers have mostly been found to be elevated in agricultural soils, the reason for this has not been explained (i.e. are these actual cells or just organic matter on the dust particle). Thus, the conclusion that CL3 is a mixed dust-bacterial type is not well-founded.

After the introduction and for the rest of the manuscript, there are numerous places where there are line breaks that do not result in a new paragraph (see Page 4, line 17. These line breaks should either be removed, or a new paragraph started to be consistent with the rest of the manuscript. There are numerous formatting choices that do not follow the style guidelines. I've highlight some of these in the specific comments below, but this list is not exhaustive.

Specific comments

Page 4, line 13. I think that reporting meteorological conditions from Carpenter et al. is not needed as there is contemporaneous meteorological data at the site, which should be reported instead.

Page 4, line 15. ms should be m s$^{-1}$.

Page 4, line 23. Was the WIBS-4M calibrated?

Page 4, line 30. There appear to be too many parentheses on this line.

Page 4, line 31. There needs to be a space before Liu.

Page 5, line 1. "WIBS-4" should be "WIBS-4A."

Page 5, line 3. There are two spaces in front of 1000.

Page 5, line 3, 4, 5, 6. There should be a space between the number value and its unit.

Page 5, line 5. The LAAP-ToF can measure both refractory and non-refractory aerosol, and presumably did so here. I would recommend removing.

Page 5, line 5. The LAAP-ToF and the WIBS measure different diameters ($D_{va}$ and $D_o$) and this information should be included here.

Page 5, line 9. "anc" should be "and."

Page 5, line 10. This sentence is not clear. Does Saharan dust influence both prior to 15 August and mid July?

Page 5, line 22. Do abbreviations (e.g. UV-LIF) need to be re-introduced if they are introduced in the abstract? I thought they did, but maybe the editors can weigh in.

Page 5, line 25. Why just the negative spectra? Isn't the LAAP-ToF capable of acquiring dual polarity spectra?

Page 6, line 32. These first two sentences are similar and should be combined.

Page 7, line 13. "Interferences" should be "interferents."

Page 7, line 15. Savage et al. (2017) found that applying the 9 sigma threshold helped reduce weakly fluorescent dust particles, but that it had no effect for soot, which needs to be discussed and accounted for in the analysis.

Page 7, line 17. What is $D_{p50}$?

Page 9, line 3. Space between "Fig." and "1a."

Page 10, line 3. The sentence starting on this line and the one following it should be combined.

Page 14, line 3. Sentences starting with "Figure X" should use the unabbreviated version (per style guidleines).

Page 16, line 2. A reference to Figure 6 should be included somewhere in this paragraph.

Table 1. Units should be given for size. In the caption, asymmetry should be un-capitalized.

Figure 1. Panels should be labelled with two parentheses around each label. I would suggest adding it in the panel in the upper left corner as it does look a little strange above. This extends to the caption. Effective diameter should not be capitalized and should be abbreviated to match figure label.

Figure 2. Labelling is not clear across plots. Caption needs to describe each panel. Panels need to follow style guidelines.

Figure 9. For the inset plot, it appears that the x-axis shows the fractional of SilicateBio. Is that fraction of all particles, or fraction of silicate particles with bio?

---

## Author Comment (AC1) · 20 Jul 2020

We appreciate the feedback and points raised. The comments regarding formatting and typos have now been implemented, alongside the reviewer's suggestions regarding changes to the figures. Where they have raised an interesting point requiring a more detailed discussion, our responses are written below. The original referee comment is written in black text with our responses written in blue. Sentences from the original draft of the manuscript are crossed through and written in red, while the new sentences are in red and have been italicised.

General Comments

1.Use of a forced trigger+9 σ threshold for WIBS measurements has been shown to reduce potential non-biological interferents mineral dust samples, but not for all particle types, e.g. soot. This is particularly important because much of the analysis in the paper depends upon the assumption that all fluorescent particles are biological (page 7, line 15). This should be addressed somehow.

> There are two main reasons why we do not believe we are observing fluorescent non-biological particles such as soot. Firstly, previous studies have demonstrated that soot can have variable levels of fluorescence. Toprak and Schnaiter, (2013) found propane flame soot to only weakly fluoresce in FL1 using 3 sigma thresholding. We would therefore not expect this type of soot to register as fluorescent when using 9 sigma thresholding. Secondly, the size of the observed fluorescent particles are larger than we would expect for soot. Toprak and Schnaiter, (2013) found generated soot to only be 0.8 µm after significant coagulation time in the NAUA chamber. Savage et al., (2017) used a mechanically dispersed dry diesel soot powder to investigate potential interferent aerosol fluorescence. They noted that this powder fluoresced above a conservative 9 sigma threshold, however, this sample aerosol was much larger than soot typically observed in the atmosphere when aerosolised (~ 1.1 µm). Savage et al., (2017) also acknowledged that fluorescent intensity is a strong function of particle size owing to surface area/volume effects and that this test soot was likely to be significantly more fluorescent than ambient diesel soot as a result. Furthermore, Savage and Huffman et al., (2018) acknowledge that more highly fluorescent soot is representative of freshly generated soot close to source, and is not representative of aged or processed soot. Ambient soot at CVAO should not be fluorescent at 9 sigma. While it is possible that soot could have internally mixed with dust and therefore become larger, this would still represent aged soot and would be less fluorescent.

2. Using two different versions of the WIBS is a weakness of the study. I think that some context on the difference between the 4A and 4M would be helpful in order to understand to how equivalent these two datasets are. Additionally, I found it difficult to tell whether the particle clustering from the two WIBS datasets was performed together or separately. If so, some comparison between the two sets of clusters would be helpful to make sure that they are really equivalent. Also, how was the WIBS-4M calibrated? The manuscript only mentions the 4A.

> The two datasets were clustered separately and a line has been added to Section 2.3 highlighting this. In regard to using two different instruments, we feel it important to note the WIBS-4A and 4M are essentially identical, with the only differences being the trigger levels and flow rates used. A detailed description of the 4A can be found in Savage et al., (2017) and of the 4M in Forde et al., (2019). As such, the fluorescence data for each instrument are comparable, although we acknowledge there can be issues even when comparing measurements from two identical models. To help reduce uncertainty here an inter-comparison of the fluorescent responses

between the two instruments was conducted using NIST (National Institute of Standards and Technology) calibration polystyrene latex (PSL) spheres. This is the same methodology as has been described by Forde et al., (2019), Savage et al., (2017) and Crawford et al., (2014).

3. More information is needed on the back-trajectory analysis. How many were released, and how often? Some context on what the Openair package analysis provides would be helpful for the reader.

Additional information has been added to clarify what the back trajectories represent. They were calculated in 3 hour intervals for all 11 months of the campaign, using monthly meteorological data files available from NOAA.

4. For the LAAP-ToF measurements, it should be noted that simply using CN- and CNO- as boolean-type markers for biological material is suspect. They are pretty non-specific and found in a number of different particle types. This is discussed in detail by Zawadowicz et al. (2017). While these markers have mostly been found to be elevated in agricultural soils, the reason for this has not been explained (i.e. are these actual cells or just organic matter on the dust particle). Thus, the conclusion that CL3 is a mixed dust-bacterial type is not well-founded.

In regard to the markers we have used to classify particles as biological, it is true that previous studies have shown these markers to occur in organic soil matter where biological components are not expected (Sodeman et al., (2005); Zawadowicz et al., (2017); Wonaschuetz et al., (2017)). However, by including data from the WIBS-4M and observing a high level of agreement with particle counts from the LAAP-ToF, we are confident what we observe is biological in nature. This is a similar approach to a study by Zawadowicz et al., (2019), except we have not used a phosphate marker as they can have an inorganic dust origin. It should be emphasised here that interferents such as soot, dust and polycyclic aromatic hydrocarbons (PAHs) were aerosolised in laboratory experiments by Savage et al., (2017), who concluded that a 9 sigma threshold would effectively remove the majority of non-biological particles. Soot is one of the few exceptions, but can be discounted for the reasons discussed above. As such, we are confident the trends in the WIBS data are not due to known interferents. This allows us to compare the biofluorescent signature data with the LAAP-ToF mass spectral signatures and due to the high level of agreement between the instruments provides a good degree of confidence that the LAAP-ToF is also observing genuine biological material.

5. After the introduction and for the rest of the manuscript, there are numerous places where there are line breaks that do not result in a new paragraph (see Page 4, line 17. These line breaks should either be removed, or a new paragraph started to be consistent with the rest of the manuscript. There are numerous formatting choices that do not follow the style guidelines. I've highlight some of these in the specific comments below, but this list is not exhaustive.

The line breaks have now been changed to be consistent throughout the manuscript. The formatting issues highlighted have also been addressed.

Specific Comments

Page 4, line 13. I think that reporting meteorological conditions from Carpenter et al. is not needed as there is contemporaneous meteorological data at the site, which should be reported instead.

Description of the meteorological conditions has been changed to more contemporaneous data.

*During this period the observatory recorded a mean temperature of 23.3◦C and median wind speeds of 6.2 ms$^{-1}$.*

Page 4, line 15. ms should be ms$^{-1}$.

ms has been changed to ms$^{-1}$.

Page 4, line 23. Was the WIBS-4M calibrated?

A line has now been added in the methodology to clarify that the WIBS-4M was calibrated. See above for detail.

*The WIBS-4M was used to observe long-term trends in bioaerosol concentrations and was regularly calibrated using National Institute of Standards and Technology (NIST) calibration polystyrene latex (PSL) spheres.*

Page 4, line 30. There appear to be too many parentheses on this line.

A parenthesis was removed.

Page 4, line 31. There needs to be a space before Liu.

A space has been added before Liu.

Page 5, line 1. "WIBS-4" should be "WIBS-4A."

WIBS-4 has been changed to WIBS-4A

Page 5, line 3. There are two spaces in front of 1000.

One of the spaces in front of 1000 has been removed.

Page 5, line 3, 4, 5, 6. There should be a space between the number value and its unit.

A space has been added between the number values and their unit.

Page 5, line 5. The LAAP-ToF can measure both refractory and non-refractory aerosol, and presumably did so here. I would recommend removing.

The line regarding measurements of refractory and non-refractory aerosol has been removed.

*The LAAP-ToF measured aerosol in the size range 0.5-2.5 µm whilst the WIBS-4A measured particles in the size range 0.5-20 µm, however the nominal cut-off of the inlet system was approximately 10 µm.*

Page 5, line 5. The LAAP-ToF and the WIBS measure different diameters (Dva and Do) and this information should be included here.

A line clarifying the $D_{va}$ and $D_o$ has been added.

*When considering size measurements, it should be noted that the LAAP-ToF measures a particle's vacuum aerodynamic diameter ($D_{va}$) while the WIBS-4A measure a particle's optical diameter ($D_o$).*

Page 5, line 9. "anc" should be "and."

'anc' has been changed to and.

Page 5, line 10. This sentence is not clear. Does Saharan dust influence both prior to 15 August and mid July?

The influence of Saharan dust on various dates has been clarified. The text now only discusses a synoptic shift on the 15th August that resulted in more easterly winds.

*They reported that dust plumes from the Sahara and sub-Saharan Africa were more frequently observed after a synoptic shift on the 15th August, when winds became more easterly.*

Page 5, line 22. Do abbreviations (e.g. UV-LIF) need to be re-introduced if they are introduced in the abstract? I thought they did, but maybe the editors can weigh in.

Abbreviations introduced in the abstract have been repeated in the main body of the text (e.g. UV-LIF).

Page 5, line 25. Why just the negative spectra? Isn't the LAAP-ToF capable of acquiring dual polarity spectra?

To the question raised about why we only use the negative spectra, this is because as noted by Zawadowicz et al., (2019), successful classification of bioaerosols can only be performed in the negative mode. In their particle analysis by laser mass spectrometry (PALMS), the phosphate ions $PO_3^-$ and $PO_2^-$, are key features used in the classification (Zawadowicz et al., 2017) and they are prominent only in the negative PALMS spectra.

Page 6, line 32. These first two sentences are similar and should be combined.

Two sentences regarding the LAAP-ToF methodology have been combined.

*As a technique, SMPS is considered qualitative or semi-quantitative. The ablation and ionisation process of particles is incomplete so that competitive ionisation and charge transfer in the vapour plume results in a strong matrix effect (Reinard and Johnston, 2008).*

Page 7, line 13. "Interferences" should be "interferents."

Interferences has been changed to interferents.

Page 7, line 15. Savage et al. (2017) found that applying the 9 sigma threshold helped reduce weakly fluorescent dust particles, but that it had no effect for soot, which needs to be discussed and accounted for in the analysis.

The discussion regarding soot can be seen above.

Page 7, line 17. What is Dp50?

$D_{p50}$ or $D_{50}$ is a standard metric used in aerosol science to describe the minimum cut-off diameter at 50% detection efficiency for a particular aerosol spectrometer. For the WIBS optical particle counter instruments this can vary depending upon the version and flow rate. For the original WIBS-3 e.g. the $D_{p50}$ was determined in the laboratory to be 0.8 µm (Gabey et al., 2010) and the WIBS-4M was also 0.8 µm (Crawford, Ruske, Topping and Gallagher, 2015). Other versions of the WIBS-4 have been shown to have a $D_{50}$ of 0.49 µm, (Healy, O'Connor and Sodeau, 2012). This has been included into the main body of the text.

*The minimum cut-off diameter at 50% detection efficiency ( $D_{p50}$ ) for this particular WIBS-4 is 0.8 µm. As such, only particles between 0.8 and 10 µm have been included in the analysis.*

Page 9, line 3. Space between "Fig." and "1a."

A space has been added between Fig and 1a.

Page 10, line 3. The sentence starting on this line and the one following it should be combined.

The two sentences regarding high fraction events have been combined.

*The 99th percentile value was a fraction of 1.1 %, with some high fraction events exceeding 1.5 % seen in Oct and May.*

Page 14, line 3. Sentences starting with "Figure X" should use the unabbreviated version (per style guidleines).

For the sentences beginning with Figure, the unabbreviated version has now been used.

Page 16, line 2. A reference to Figure 6 should be included somewhere in this paragraph.

A reference to Figure 6 has been included in the main body of the text.

*It can be seen in Figure. 6 that all of the dust samples dispersed through the WIBS-4M show relatively weak fluorescence across all three channels, but are on average more fluorescent than clusters 2 and 3.*

Table 1. Units should be given for size. In the caption, asymmetry should be un-capitalized.

Table headings have now been given appropriate units and the asymmetry factor is no longer capitalised.

Figure 1. Panels should be labelled with two parentheses around each label. I would suggest adding it in the panel in the upper left corner as it does look a little strange above. This extends to the caption. Effective diameter should not be capitalized and should be abbreviated to match figure label.

Labels have been changed/moved as suggested for Figure 1. Captions have also been changed accordingly.

Figure 2. Labelling is not clear across plots. Caption needs to describe each panel. Panels need to follow style guidelines.

The labels have been changed in the same way as Figure 1. The caption has been expanded to include descriptions of each panel.

*Figure 2. Solid lines represent the median fluorescence of particles as a function of size, instrument channel and cluster. Dotted lines represent 95th percentile values. 2.5 µm bins have been used for all fluorescence traces. Panels (a-d) represent clusters 1-4 respectively, while panel (e). represents the normalised size distribution of fluorescent particles across all clusters, using 0.5 µm bins. Panel (f) represents the average across all fluorescent particles.*

Figure 9. For the inset plot, it appears that the x-axis shows the fractional of SilicateBio. Is that fraction of all particles, or fraction of silicate particles with bio?

The x-axis for the inset plot is normalised to the maximum observed concentration of bio-silicate dust. This has now been emphasised in the caption to remove any ambiguity.

*Figure 9.Time series of WIBS Cl3 and silicate dust filtered for Bio markers using 20 minute averaged number counts normalised by the maximum observed concentration for each instrument. Values for the inset plot are also normalised to the maximum observed concentration.*

**References**

Crawford, I., Robinson, N., Flynn, M., Foot, V., Gallagher, M., Huffman, J., Stanley, W. and Kaye, P., 2014. Characterisation of bioaerosol emissions from a Colorado pine forest: results from the BEACHON-RoMBAS experiment. *Atmospheric Chemistry and Physics*, 14(16), pp.8559-8578.

Crawford, I., Ruske, S., Topping, D. and Gallagher, M., 2015. Evaluation of hierarchical agglomerative cluster analysis methods for discrimination of primary biological aerosol. *Atmospheric Measurement Techniques*, 8(11), pp.4979-4991.

Forde, E., Gallagher, M., Walker, M., Foot, V., Attwood, A., Granger, G., Sarda-Estève, R., Stanley, W., Kaye, P. and Topping, D., 2019. Intercomparison of Multiple UV-LIF Spectrometers Using the Aerosol Challenge Simulator. *Atmosphere*, 10(12), p.797.

Gabey, A., Gallagher, M., Whitehead, J., Dorsey, J., Kaye, P. and Stanley, W., 2010. Measurements and comparison of primary biological aerosol above and below a tropical forest canopy using a dual channel fluorescence spectrometer. *Atmospheric Chemistry and Physics*, 10(10), pp.4453-4466.

Healy, D., O'Connor, D. and Sodeau, J., 2012. Measurement of the particle counting efficiency of the "Waveband Integrated Bioaerosol Sensor" model number 4 (WIBS-4). *Journal of Aerosol Science*, 47, pp.94-99.

Savage, N., Krentz, C., Könemann, T., Han, T., Mainelis, G., Pöhlker, C. and Huffman, J., 2017. Systematic characterization and fluorescence threshold strategies for the wideband integrated bioaerosol sensor (WIBS) using size-resolved biological and interfering particles. *Atmospheric Measurement Techniques*, 10(11), pp.4279-4302.

Savage, N. and Huffman, J., 2018. Evaluation of a hierarchical agglomerative clustering method applied to WIBS laboratory data for improved discrimination of biological particles by comparing data preparation techniques. *Atmospheric Measurement Techniques*, 11(8), pp.4929-4942.

Sodeman, D., Toner, S. and Prather, K., 2005. Determination of Single Particle Mass Spectral Signatures from Light-Duty Vehicle Emissions. *Environmental Science & Technology*, 39(12), pp.4569-4580.

Toprak, E. and Schnaiter, M., 2013. Fluorescent biological aerosol particles measured with the Waveband Integrated Bioaerosol Sensor WIBS-4: laboratory tests combined with a one year field study. *Atmospheric Chemistry and Physics*, 13(1), pp.225-243.

Wonaschuetz, A., Kallinger, P., Szymanski, W. and Hitzenberger, R., 2017. Chemical composition of radiolytically formed particles using single-particle mass spectrometry. *Journal of Aerosol Science*, 113, pp.242-249.

Zawadowicz, M., Froyd, K., Murphy, D. and Cziczo, D., 2017. Improved identification of primary biological aerosol particles using single-particle mass spectrometry. *Atmospheric Chemistry and Physics*, 17(11), pp.7193-7212.

Zawadowicz, M., Froyd, K., Perring, A., Murphy, D., Spracklen, D., Heald, C., Buseck, P. and Cziczo, D., 2019. Model-measurement consistency and limits of bioaerosol abundance over the continental United States. *Atmospheric Chemistry and Physics*, 19(22), pp.13859-13870.

---

## Author Response (AR1)

We appreciate the feedback and points raised. The comments regarding formatting and typos have now been implemented, alongside the reviewer's suggestions regarding changes to the figures. Where they have raised an interesting point requiring a more detailed discussion, our responses are written below. The original referee comment is written in black text with our responses written in blue. Sentences from the original draft of the manuscript are crossed through and written in red, while the new sentences are in red and have been italicised.

General Comments

1.Use of a forced trigger+9 σ threshold for WIBS measurements has been shown to reduce potential non-biological interferents mineral dust samples, but not for all particle types, e.g. soot. This is particularly important because much of the analysis in the paper depends upon the assumption that all fluorescent particles are biological (page 7, line 15). This should be addressed somehow.

> There are two main reasons why we do not believe we are observing fluorescent non-biological particles such as soot. Firstly, previous studies have demonstrated that soot can have variable levels of fluorescence. Toprak and Schnaiter, (2013) found propane flame soot to only weakly fluoresce in FL1 using 3 sigma thresholding. We would therefore not expect this type of soot to register as fluorescent when using 9 sigma thresholding. Secondly, the size of the observed fluorescent particles are larger than we would expect for soot. Toprak and Schnaiter, (2013) found generated soot to only be 0.8 μm after significant coagulation time in the NAUA chamber. Savage et al., (2017) used a mechanically dispersed dry diesel soot powder to investigate potential interferent aerosol fluorescence. They noted that this powder fluoresced above a conservative 9 sigma threshold, however, this sample aerosol was much larger than soot typically observed in the atmosphere when aerosolised (~ 1.1 μm). Savage et al., (2017) also acknowledged that fluorescent intensity is a strong function of particle size owing to surface area/volume effects and that this test soot was likely to be significantly more fluorescent than ambient diesel soot as a result. Furthermore, Savage and Huffman et al., (2018) acknowledge that more highly fluorescent soot is representative of freshly generated soot close to source, and is not representative of aged or processed soot. Ambient soot at CVAO should not be fluorescent at 9 sigma. While it is possible that soot could have internally mixed with dust and therefore become larger, this would still represent aged soot and would be less fluorescent.

2. Using two different versions of the WIBS is a weakness of the study. I think that some context on the difference between the 4A and 4M would be helpful in order to understand to how equivalent these two datasets are. Additionally, I found it difficult to tell whether the particle clustering from the two WIBS datasets was performed together or separately. If so, some comparison between the two sets of clusters would be helpful to make sure that they are really equivalent. Also, how was the WIBS-4M calibrated? The manuscript only mentions the 4A.

> The two datasets were clustered separately and a line has been added to Section 2.3 highlighting this. In regard to using two different instruments, we feel it important to note the WIBS-4A and 4M are essentially identical, with the only differences being the trigger levels and flow rates used. A detailed description of the 4A can be found in Savage et al., (2017) and of the 4M in Forde et al., (2019). As such, the fluorescence data for each instrument are comparable, although we acknowledge there can be issues even when comparing measurements from two identical models. To help reduce uncertainty here an inter-comparison of the fluorescent responses

between the two instruments was conducted using NIST (National Institute of Standards and Technology) calibration polystyrene latex (PSL) spheres. This is the same methodology as has been described by Forde et al., (2019), Savage et al., (2017) and Crawford et al., (2014).

3. More information is needed on the back-trajectory analysis. How many were released, and how often? Some context on what the Openair package analysis provides would be helpful for the reader.

Additional information has been added to clarify what the back trajectories represent. They were calculated in 3 hour intervals for all 11 months of the campaign, using monthly meteorological data files available from NOAA.

4. For the LAAP-ToF measurements, it should be noted that simply using CN- and CNO- as boolean-type markers for biological material is suspect. They are pretty non-specific and found in a number of different particle types. This is discussed in detail by Zawadowicz et al. (2017). While these markers have mostly been found to be elevated in agricultural soils, the reason for this has not been explained (i.e. are these actual cells or just organic matter on the dust particle). Thus, the conclusion that CL3 is a mixed dust-bacterial type is not well-founded.

In regard to the markers we have used to classify particles as biological, it is true that previous studies have shown these markers to occur in organic soil matter where biological components are not expected (Sodeman et al., (2005); Zawadowicz et al., (2017); Wonaschuetz et al., (2017)). However, by including data from the WIBS-4M and observing a high level of agreement with particle counts from the LAAP-ToF, we are confident what we observe is biological in nature. This is a similar approach to a study by Zawadowicz et al., (2019), except we have not used a phosphate marker as they can have an inorganic dust origin. It should be emphasised here that interferents such as soot, dust and polycyclic aromatic hydrocarbons (PAHs) were aerosolised in laboratory experiments by Savage et al., (2017), who concluded that a 9 sigma threshold would effectively remove the majority of non-biological particles. Soot is one of the few exceptions, but can be discounted for the reasons discussed above. As such, we are confident the trends in the WIBS data are not due to known interferents. This allows us to compare the biofluorescent signature data with the LAAP-ToF mass spectral signatures and due to the high level of agreement between the instruments provides a good degree of confidence that the LAAP-ToF is also observing genuine biological material.

5. After the introduction and for the rest of the manuscript, there are numerous places where there are line breaks that do not result in a new paragraph (see Page 4, line 17. These line breaks should either be removed, or a new paragraph started to be consistent with the rest of the manuscript. There are numerous formatting choices that do not follow the style guidelines. I've highlight some of these in the specific comments below, but this list is not exhaustive.

The line breaks have now been changed to be consistent throughout the manuscript. The formatting issues highlighted have also been addressed.

Specific Comments

Page 4, line 13. I think that reporting meteorological conditions from Carpenter et al. is not needed as there is contemporaneous meteorological data at the site, which should be reported instead.

Description of the meteorological conditions has been changed to more contemporaneous data.

*During this period the observatory recorded a mean temperature of 23.3◦C and median wind speeds of 6.2 ms$^{-1}$.*

Page 4, line 15. ms should be ms$^{-1}$.

ms has been changed to ms$^{-1}$.

Page 4, line 23. Was the WIBS-4M calibrated?

A line has now been added in the methodology to clarify that the WIBS-4M was calibrated. See above for detail.

*The WIBS-4M was used to observe long-term trends in bioaerosol concentrations and was regularly calibrated using National Institute of Standards and Technology (NIST) calibration polystyrene latex (PSL) spheres.*

Page 4, line 30. There appear to be too many parentheses on this line.

A parenthesis was removed.

Page 4, line 31. There needs to be a space before Liu.

A space has been added before Liu.

Page 5, line 1. "WIBS-4" should be "WIBS-4A."

WIBS-4 has been changed to WIBS-4A

Page 5, line 3. There are two spaces in front of 1000.

One of the spaces in front of 1000 has been removed.

Page 5, line 3, 4, 5, 6. There should be a space between the number value and its unit.

A space has been added between the number values and their unit.

Page 5, line 5. The LAAP-ToF can measure both refractory and non-refractory aerosol, and presumably did so here. I would recommend removing.

The line regarding measurements of refractory and non-refractory aerosol has been removed.

*The LAAP-ToF measured aerosol in the size range 0.5-2.5 µm whilst the WIBS-4A measured particles in the size range 0.5-20 µm, however the nominal cut-off of the inlet system was approximately 10 µm.*

Page 5, line 5. The LAAP-ToF and the WIBS measure different diameters (Dva and Do) and this information should be included here.

A line clarifying the $D_{va}$ and $D_o$ has been added.

*When considering size measurements, it should be noted that the LAAP-ToF measures a particle's vacuum aerodynamic diameter ($D_{va}$) while the WIBS-4A measure a particle's optical diameter ($D_o$).*

Page 5, line 9. "anc" should be "and."

'anc' has been changed to and.

Page 5, line 10. This sentence is not clear. Does Saharan dust influence both prior to 15 August and mid July?

The influence of Saharan dust on various dates has been clarified. The text now only discusses a synoptic shift on the 15th August that resulted in more easterly winds.

*They reported that dust plumes from the Sahara and sub-Saharan Africa were more frequently observed after a synoptic shift on the 15th August, when winds became more easterly.*

Page 5, line 22. Do abbreviations (e.g. UV-LIF) need to be re-introduced if they are introduced in the abstract? I thought they did, but maybe the editors can weigh in.

Abbreviations introduced in the abstract have been repeated in the main body of the text (e.g. UV-LIF).

Page 5, line 25. Why just the negative spectra? Isn't the LAAP-ToF capable of acquiring dual polarity spectra?

To the question raised about why we only use the negative spectra, this is because as noted by Zawadowicz et al., (2019), successful classification of bioaerosols can only be performed in the negative mode. In their particle analysis by laser mass spectrometry (PALMS), the phosphate ions $PO_3^-$ and $PO_2^-$, are key features used in the classification (Zawadowicz et al., 2017) and they are prominent only in the negative PALMS spectra.

Page 6, line 32. These first two sentences are similar and should be combined.

Two sentences regarding the LAAP-ToF methodology have been combined.

*As a technique, SMPS is considered qualitative or semi-quantitative. The ablation and ionisation process of particles is incomplete so that competitive ionisation and charge transfer in the vapour plume results in a strong matrix effect (Reinard and Johnston, 2008).*

Page 7, line 13. "Interferences" should be "interferents."

Interferences has been changed to interferents.

Page 7, line 15. Savage et al. (2017) found that applying the 9 sigma threshold helped reduce weakly fluorescent dust particles, but that it had no effect for soot, which needs to be discussed and accounted for in the analysis.

The discussion regarding soot can be seen above.

Page 7, line 17. What is Dp50?

$D_{p50}$ or $D_{50}$ is a standard metric used in aerosol science to describe the minimum cut-off diameter at 50% detection efficiency for a particular aerosol spectrometer. For the WIBS optical particle counter instruments this can vary depending upon the version and flow rate. For the original WIBS-3 e.g. the $D_{p50}$ was determined in the laboratory to be 0.8 µm (Gabey et al., 2010) and the WIBS-4M was also 0.8 µm (Crawford, Ruske, Topping and Gallagher, 2015). Other versions of the WIBS-4 have been shown to have a $D_{50}$ of 0.49 µm, (Healy, O'Connor and Sodeau, 2012). This has been included into the main body of the text.

*The minimum cut-off diameter at 50% detection efficiency ( $D_{p50}$ ) for this particular WIBS-4 is 0.8 µm. As such, only particles between 0.8 and 10 µm have been included in the analysis.*

Page 9, line 3. Space between "Fig." and "1a."

A space has been added between Fig and 1a.

Page 10, line 3. The sentence starting on this line and the one following it should be combined.

The two sentences regarding high fraction events have been combined.

*The 99th percentile value was a fraction of 1.1 %, with some high fraction events exceeding 1.5 % seen in Oct and May.*

Page 14, line 3. Sentences starting with "Figure X" should use the unabbreviated version (per style guidlienes).

For the sentences beginning with Figure, the unabbreviated version has now been used.

Page 16, line 2. A reference to Figure 6 should be included somewhere in this paragraph.

A reference to Figure 6 has been included in the main body of the text.

*It can be seen in Figure. 6 that all of the dust samples dispersed through the WIBS-4M show relatively weak fluorescence across all three channels, but are on average more fluorescent than clusters 2 and 3.*

Table 1. Units should be given for size. In the caption, asymmetry should be un-capitalized.

Table headings have now been given appropriate units and the asymmetry factor is no longer capitalised.

Figure 1. Panels should be labelled with two parentheses around each label. I would suggest adding it in the panel in the upper left corner as it does look a little strange above. This extends to the caption. Effective diameter should not be capitalized and should be abbreviated to match figure label.

Labels have been changed/moved as suggested for Figure 1. Captions have also been changed accordingly.

Figure 2. Labelling is not clear across plots. Caption needs to describe each panel. Panels need to follow style guidelines.

The labels have been changed in the same way as Figure 1. The caption has been expanded to include descriptions of each panel.

*Figure 2. Solid lines represent the median fluorescence of particles as a function of size, instrument channel and cluster. Dotted lines represent 95th percentile values. 2.5 µm bins have been used for all fluorescence traces. Panels (a-d) represent clusters 1-4 respectively, while panel (e). represents the normalised size distribution of fluorescent particles across all clusters, using 0.5 µm bins. Panel (f) represents the average across all fluorescent particles.*

Figure 9. For the inset plot, it appears that the x-axis shows the fractional of SilicateBio. Is that fraction of all particles, or fraction of silicate particles with bio?

The x-axis for the inset plot is normalised to the maximum observed concentration of bio-silicate dust. This has now been emphasised in the caption to remove any ambiguity.

*Figure 9.Time series of WIBS Cl3 and silicate dust filtered for Bio markers using 20 minute averaged number counts normalised by the maximum observed concentration for each instrument. Values for the inset plot are also normalised to the maximum observed concentration.*

**Compare Results**

| Old File: | | New File: |
|---|---|---|
| **CVAOpaperasSubmitted.pdf** | versus | **CVAOResponsePaper.pdf** |
| **29 pages (2.25 MB)** | | **28 pages (2.17 MB)** |
| 16/06/2020 12:30:12 | | 03/08/2020 11:53:41 |

**Total Changes**

**243**

**Content**

129    Replacements

51    Insertions

49    Deletions

**Styling and Annotations**

7    Styling

7    Annotations

Go to First Change (page 1)

**Quantifying Bioaerosol Concentrations in Dust Clouds through Online UV-LIF and Mass Spectrometry Measurements at the Cape Verde Atmospheric Observatory**

Douglas Morrison,[1], Ian Crawford,[1], Nicholas Marsden,[1], Michael Flynn,[1], Katie Read,[2], Luis Neves,[2], Virginia Foot,[4], Paul Kaye,[3], Warren Stanley,[3], Hugh Coe,[1], David Topping,[1], and Martin Gallagher.[1]

[1]Department of Earth and Environmental Science, University of Manchester, Brunswick St, Manchester, M13 9PS
[2]Wolfson Atmospheric Chemistry Laboratory, University of York, York, YO10 5DD
[3]Science and Technology Research Institute, University of Hertfordshire, Hatfield, U.K.
[4]Defence Science and Technology Laboratory, Salisbury, U.K

**Correspondence:** Douglas Morrison (douglas.morrison@manchester.ac.uk)

**Abstract.** Observations of the long-range transport of biological particles in the tropics via dust vectors are now seen as fundamental to the understanding of many global atmosphere-oceanic biogeochemical cycles, changes in air quality, human health, ecosystem impacts, and climate. However, there is a lack of long-term measurements quantifying their presence in such conditions. Here we present annual observations of bioaerosol concentrations based on online ultraviolet light induced fluorescence (UV-LIF) spectrometry from the global WMO/Global Atmospheric Watch (GAW) observatory on Sao Vicente Cape Verde Atmospheric Observatory. We observe the expected strong seasonal changes in absolute concentrations of bioaerosols with significant enhancements during winter due to the strong island inflow of airmass, originating from the African continent. Monthly median bioaerosol concentrations as high as $45 \text{ L}^{-1}$ were found with 95th percentile values exceeding $130 \text{ L}^{-1}$ during strong dust events. However, in contrast the relative fraction of bioaerosol numbers compared to total dust number concentration shows little seasonal variation. Mean bioaerosol contributions accounted for $0.4 \pm 0.2\%$ of total coarse aerosol concentrations, only rarely exceeding 1% during particularly strong events under appropriate conditions. Although enhancements in the median bioaerosol fraction do occur in winter, they also occur at other times of the year, likely due to the enhanced Aeolian activity driving dust events at this time from different sources. We hypothesise that this indicates the relative contribution of bioaerosol material in dust transported across the tropical Atlantic throughout the year is relatively uniform, comprised mainly of mixtures of dust and bacteria and/or bacterial fragments. We argue that this hypothesis is supported from analysis of measurements also at Cape Verde just prior to the long-term monitoring experiment where UV-LIF single particle measurements were compared with Laser Ablation Aerosol Particle Time of Flight mass spectrometer (LAAP-ToF) measurements. These clearly show a very high correlation between particles with mixed bio-silicate mass spectral signatures and UV-LIF bio-fluorescent signatures suggesting the bioaerosol concentrations are dominated by these mixtures. These observations should assist with constraining bioaerosol concentrations for tropical Global Climate Model (GCM) simulations. Note here we use the term "bioaerosol" to include mixtures of dust and bacterial material.

*Copyright statement.* The author's copyright for this publication is transferred to the University of Manchester.

[revised manuscript text omitted]

---

## Author Response (AR2)

For item 1, I think it would be appropriate to add a brief discussion about the potential for non-biological particles to be labeled as fluorescent even with a 9-sigma threshold and your reasoning for why that is unlikely to be a problem here.

We have added a paragraph to the end of Section 4.2 explaining why soot and other non-biological particles are not thought to be influencing our observations. This is written below.

*Although the 9 sigma threshold we have used should eliminate weakly fluorescent non-biological particles, the potential for more highly fluorescent particles to act as interferants should be discussed. Soot is one example, with previous studies having observed higher fluorescence than is typically seen for non-biological particles. Despite this, there are multiple reasons that we do not believe interferants are contributing to particle concentrations. Firstly, studies that found soot to fluoresce above their thresholds had typically only done so when using 3 sigma thresholding. Toprak and Schnaiter, (2013) found propane flame soot to only weakly fluoresce in Fl1 at this threshold, and so we would not expect it to be considered fluorescent at a more conservative 9 sigma thresholding. Secondly, the size of the observed fluorescent particles are larger than we would expect for soot. Toprak and Schnaiter, (2013) found generated soot to only be 0.8 µm after significant coagulation time in the NAUA chamber, while Savage et al., (2017) used a mechanically dispersed dry diesel soot powder to investigate potential interferent aerosol fluorescence. They noted that this powder fluoresced above a conservative 9 sigma threshold, but this sample aerosol was much larger than soot typically observed in the atmosphere when aerosolised (~ 1.1 µm). Savage et al., (2017) also acknowledged that fluorescent intensity is a strong function of particle size owing to surface area/volume effects and that this test soot was likely to be significantly more fluorescent than ambient diesel soot as a result. Furthermore, Savage and Huffman et al., (2018) acknowledge that more highly fluorescent soot is representative of freshly generated soot close to source, and is not representative of aged or processed soot. Ambient soot at CVAO should not be fluorescent at 9 sigma. While it is possible that soot could have internally mixed with dust and therefore become larger, this would still represent aged soot and would be less fluorescent.*

I would also recommend adding text about point 2, regarding the potential effect of using data from two different models of WIBS instruments

In Section 2.1.1 we have inserted some text discussing the use of two models of WIBS.

*Calibration of the LAAP-ToF was performed with pure hematite samples (Liu et al., 2018), whilst both the WIBS-4A andWIBS-4M were calibrated using NIST latex calibration beads and fluorescent glass beads, e.g. Crawford et al. (2015). It should be emphasised here that the WIBS-4A and 4M are almost identical instruments, with the only differences being the trigger levels and flow rates used. A detailed description of the 4A can be found in Savage et al. (2017) and of the 4M in Forde et al.(2019). As such, the fluorescence data for each instrument are comparable, although we acknowledge there can be issues even when comparing measurements from two identical models. An inter-comparison of the fluorescent responses between instruments when using the NIST calibration particles help affirm the instrument's similarities. This is the same methodology as has been described by Forde et al. (2019), Savage et al. (2017) and Crawford et al. (2014).*

With regard to item 4 (the issue of the non-specificity of the mass spec markers for biological particles), I can see your point about how the higher threshold, based on the Savage paper, should do a better job of rejecting non-biological things. However, the fraction of particles that you are identifying as biological is very small (generally <1%) and I'm not sure that the graphs in Savage et al., are fully conclusive at that level. Even if HULIS is mostly rejected at the 9-sigma threshold, it seems that just a few misidentified particles could swing your numbers quite a bit and that cluster is just barely above the threshold. Did you check whether those mass spec markers were correlated with dust or biomass emissions? If they are that might indicate that a tiny fraction is bleeding through even at the higher threshold. At a minimum I think a little bit of discussion is warranted.

> When dealing with such small percentages we agree that even small errors in classification can produce significant swings in concentrations. We believe the close correlation with particle counts for the LAAP-ToF's 'bio-silicate' class offers the strongest evidence that bleeding is not occurring, as interferant particles would have a different mass spectral profile. The role of interferants can be further discounted when considering other particle properties, for example the larger size of our biological fraction. A paragraph discussing this has been added to the end of Section 4.2.

> *We also acknowledge the fraction identified as biological is small (<1%) and that concentrations would consequently be significantly affected by even minor errors in the classification of particle types. However, if a fraction of non-biological particles were 'bleeding' through and influencing our concentrations, their mass spectral signatures would differ from our 'bio-silicate' class. As there is a close correlation between the bio-silicate particle counts and our fluorescent fraction, we do not believe that bleeding is significantly changing our observations. More studies comparing such a technique may elucidate the degree to which bleeding occurs, but we believe our study provides a good first estimate of bioaerosol concentrations in this region. As discussed by Savage et al., (2017), UV-LIF results should be considered uniquely in all situations with appreciation of possible influences. We are confident that many common interferant particles such as soot can be further discounted when evaluating properties such as particle size, as well as an appreciation for modelled back trajectories and identified source regions.*